# Deficiency in coatomer complex I causes aberrant activation of STING signalling

Annemarie Steiner[1,2,3], Katja Hrovat-Schaale [1,2], Ignazia Prigione[4], Chien-Hsiung Yu [1,2],
Pawat Laohamonthonkul[1,2], Cassandra R. Harapas[1,2], Ronnie Ren Jie Low [2,5], Dominic De Nardo [1,6],
Laura F. Dagley [2,7], Michael J. Mlodzianoski [8], Kelly L. Rogers [8], Thomas Zillinger [9,10],
Gunther Hartmann [9,11], Michael P. Gantier[12,13], Marco Gattorno[4], Matthias Geyer [3], Stefano Volpi [4,14],
Sophia Davidson[1,2,15] & Seth L. Masters [1,2,15✉]

Coatomer complex I (COPI) mediates retrograde vesicular trafficking from Golgi to the endoplasmic reticulum (ER) and within Golgi compartments. Deficiency in subunit alpha causes COPA syndrome and is associated with type I IFN signalling, although the upstream innate immune sensor involved was unknown. Using in vitro models we find aberrant activation of the STING pathway due to deficient retrograde but probably not intra-Golgi transport. Further we find the upstream cytosolic DNA sensor cGAS as essentially required to drive type I IFN signalling. Genetic deletion of COPI subunits COPG1 or COPD similarly induces type I IFN activation in vitro, which suggests that inflammatory diseases associated with mutations in other COPI subunit genes may exist. Finally, we demonstrate that inflammation in COPA syndrome patient peripheral blood mononuclear cells and COPI-deficient cell lines is ameliorated by treatment with the small molecule STING inhibitor H-151, suggesting targeted inhibition of the cGAS/STING pathway as a promising therapeutic approach.

[1] Inflammation Division, The Walter and Eliza Hall Institute of Medical Research, Parkville, VIC 3052, Australia. [2] Department of Medical Biology, University of Melbourne, Parkville, VIC 3010, Australia. [3] Institute of Structural Biology, University Hospital Bonn, 53127 Bonn, Germany. [4] Centre for Autoinflammatory Diseases and Primary Immunodeficiencies, IRCCS Istituto Giannina Gaslini, 16147 Genoa, Italy. [5] Personalised Oncology Division, The Walter and Eliza Hall Institute of Medical Research, Parkville, VIC 3052, Australia. [6] Department of Anatomy and Developmental Biology, Monash Biomedicine Discovery Institute, Monash University, Clayton, VIC 3168, Australia. [7] Advanced Technology and Biology, The Walter and Eliza Hall Institute of Medical Research, Parkville, VIC 3052, Australia. [8] Center for Dynamic Imaging, The Walter and Eliza Hall Institute of Medical Research, Parkville, VIC 3052, Australia. [9] Institute of Clinical Chemistry and Clinical Pharmacology, University Hospital Bonn, 53127 Bonn, Germany. [10] Institute of Immunology, Philipps-University Marburg, BMFZ, 35043 Marburg, Germany. [11] German Centre for Infection Research (DZIF), partner site Bonn-Cologne, 53127 Bonn, Germany. [12] Centre for Innate Immunity and Infectious Diseases, Hudson Institute of Medical Research, Clayton, VIC 3168, Australia. [13] Department of Molecular and Translational Science, Monash University, Clayton, VIC 3168, Australia. [14] University of Genoa, 16126 Genoa, Italy. [15]These authors have contributed equally: Sophia Davidson, Seth L. Masters. ✉email: masters@wehi.edu.au

COPA syndrome is a recently identified rare disorder, involving complex pathology with dysregulation of the innate and adaptive immune system[1]. The disease is caused by autosomal dominant mutations in the *COPA* gene, which encodes the α−subunit (COPα, COPA) of the coatomer complex I (COPI). COPI mediates retrograde trafficking of cargo proteins from Golgi to the endoplasmic reticulum (ER) and within cis-Golgi compartments[2]. To date, more than 17 families with mutations in *COPA* have been identified worldwide[3,4]. Clinically, symptoms occur with varying severity, including interstitial lung disease with or without pulmonary haemorrhage, inflammatory arthritis, immune-mediated kidney disease and autoantibodies, which develop in the majority of patients[1].

Disease onset occurs during childhood or early adulthood with incomplete penetrance, leaving some individuals unaffected, despite carrying the mutation[1]. Interestingly, across all reported COPA syndrome patients, a total of 11 missense mutations have been identified, almost all located within exon 8 and 9 of the *COPA* gene, which translates into a 14 amino acid stretch (aa.230-243) within the WD40 domain of the COPA protein[3,4]. Being highly conserved between species and implicated in protein-protein interactions, COPA mutants were shown to have impaired binding efficiency to dilysine-motif-containing cargo proteins, therefore causing defective retrograde transport[1]. As a consequence, ER stress, activation of the unfolded protein response (UPR) and nuclear factor kappa B (NF-κB) pathway activation were suggested pathomechanisms, which have already been linked to lung disease and autoimmunity in other studies[5].

Furthermore, Volpi and colleagues were the first to describe elevated transcription levels of type I interferons (IFNs) and interferon-stimulated genes (ISGs) in peripheral blood of COPA syndrome patients, suggesting a role of type I IFN signalling and a dysregulated innate immune response in disease pathogenesis[6]. This finding is supported by three recent case reports showing the therapeutic benefit of Janus kinase (JAK) 1/2 inhibition in COPA syndrome patients[7–9]. However, the innate immune sensor, as well as the molecular mechanisms underlying COPA syndrome pathogenesis remained unclear.

Due to the involvement of COPA in intracellular trafficking, and its subcellular localization, we hypothesised COPA could play a possible role in STING (stimulator of IFN gene, also known as MITA, MPYS and ERIS)[10–13] pathway regulation. Inactive STING forms homodimers that localize to the ER membrane and has been identified as a signalling molecule downstream of multiple intracellular nucleotide sensors including cyclic GMP-AMP synthase (cGAS)[14,15]. cGAS recognizes cytosolic double-stranded DNA independent of its sequence as a danger-associated molecular pattern (DAMP) and synthesises 2'3'-cGAMP, a second messenger detected by STING[16]. 2'3'-cGAMP binding results in a conformational change in STING, which enables translocation to the Golgi apparatus[17]. Downstream signalling occurs via IκB kinase epsilon (IKKε)- and TANK-binding-kinase 1 (TBK1)-mediated activation of transcription factors NF-κB and IFN regulatory factor 3 (IRF3)[18]. Translocation of these transcription factors to the nucleus then results in gene expression of proinflammatory cytokines and type I and III IFNs (IFNαβ and IFNλ)[19,20].

Identifying the innate immune pathway activated in COPA syndrome will provide valuable insights in disease pathology leading to beneficial pharmacological intervention for this inflammatory condition.

## Results

**Generation of cellular models for COPA syndrome.** Aiming to study the protein function of COPA in the context of innate immune signalling, we used CRISPR/Cas9 genome editing to delete *COPA* in human monocytic THP-1 and epithelial HeLa cell lines. This approach was based on the assumption that the reduced target protein binding efficiency of loss of function mutations identified in COPA syndrome patients can be mimicked by the reduction of wild-type (WT) COPA protein levels. Using single guide (sg) RNAs targeting different exons of the *COPA* gene under a Doxycycline (Dox)-inducible promoter, we generated three COPA-deficient THP-1 cells lines. Interestingly, the reduction of COPA protein levels coincided with spontaneous phosphorylation of the transcription factor signal transducer and activator of transcription 1 (STAT1), which is downstream of type I and III IFN signalling (Fig. 1a)[21,22]. Using this experimental approach, complete deletion of COPA could not be achieved long-term because COPA is essential for cell survival[23]. To circumvent this, we performed Dox-induced COPA deletion prior to every experiment and proceeded to use the THP-1 cell line with sgRNA 1 (subsequently termed COPA^deficient) as it demonstrated the greatest reduction in COPA levels and concomitantly increased STAT1 phosphorylation (pSTAT1) (Fig. 1a).

Elevated proinflammatory cytokine gene expression levels (*IL1B, IL6, IL12, IL4, IL23*) have previously been described in B cell lines derived from COPA syndrome patients and a type I IFN gene signature was reported in patient peripheral blood mononuclear cells (PBMCs)[1,6]. Similarly, qRT-PCR analysis of our COPA^deficient THP-1 cells revealed increased mRNA expression levels of proinflammatory cytokines (*TNF* and *IL6*), the type I IFN subtypes *IFNA1* and *IFNB1* as well as ISGs, such as *ISG15*, *IFIT1* and *MX1* (Fig. 1b). Furthermore, increased protein levels of IFNβ and the IFN-induced chemokine, CXCL10 [24,25] were measured by ELISA in supernatants from COPA^deficient THP-1 cells (Fig. 1c).

In order to independently confirm these findings in a different cell line, we used the same experimental approach and sgRNA1 to delete *COPA* in epithelial HeLa cells (Fig. 1d). Similar to THP-1 cells, the reduction of COPA protein levels in HeLa cells resulted in increased transcription of proinflammatory cytokines and type I IFN-induced genes (Fig. 1e), as well as elevated baseline phosphorylation of TBK1 (pTBK1), a signalling molecule downstream of several pattern recognition receptors (Fig. 1d)[19,26]. Although silencing of the cGAS/STING pathway due to immortalisation with viral oncogenes in HeLa cells has been reported[27], we confirmed pathway activity in this particular cell line used here by stimulation with synthetic DNA analogues (Supplementary Fig. 1). Therefore, we have successfully generated COPA^deficient THP-1 and HeLa cell lines that model inflammatory manifestations observed in COPA syndrome and present a useful tool to investigate the molecular mechanisms underlying this disease.

**Inflammatory signalling in COPA syndrome model cell lines is driven by STING pathway overactivation.** To identify the innate immune sensor that is driving the inflammatory response in our in vitro model of COPA syndrome, we genetically co-deleted several candidate innate immune receptors in COPA^deficient THP-1 cells (Supplementary Fig. 2). Using a hypothesis-driven approach, we excluded the involvement of inflammasome sensor NLRP3 (Supplementary Fig. 2a), cytoplasmic RNA sensor protein kinase R (PKR) (Supplementary Fig. 2b) and RNA-sensors retinoic acid-inducible gene-I (RIG-I) and melanoma differentiation-associated protein 5 (MDA5) by deletion of the shared adaptor protein mitochondrial antiviral-signalling protein (MAVS) (Supplementary Fig. 2c). Furthermore, deletion of Unc93 homolog B1 (UNC93B1), an adaptor protein essential for stability and signalling of endosomal Toll-like receptors (TLRs)

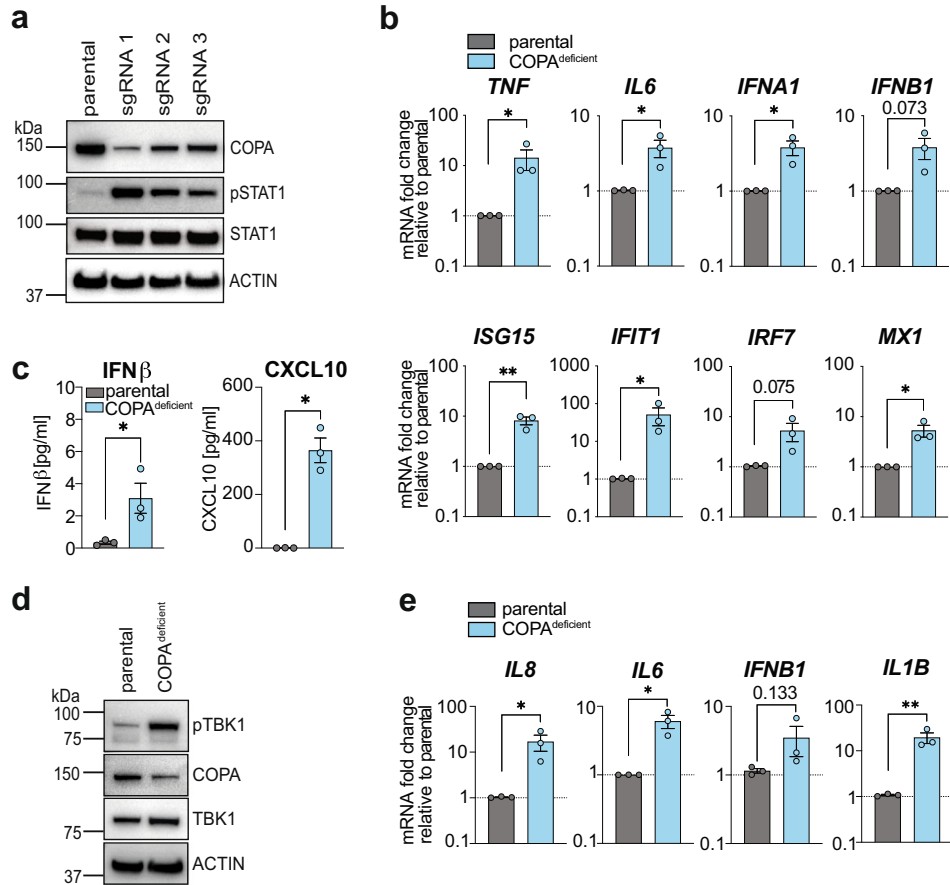

**Fig. 1 Cell line models for COPA syndrome. a** CRISPR/Cas9 gene editing technology was employed to generate COPA^deficient THP-1 cell lines. Three different single guide (sg) RNAs targeting exon 5 (sgRNA 1), exon 1 (sgRNA 2) or exon 7 (sgRNA 3) of the *COPA* gene were used. Baseline protein levels of COPA and phosphorylated STAT1 (pSTAT1) were assessed by immunoblotting, using ACTIN as a loading control. Representative result of $n = 3$ independent experiments. **b** qRT-PCR analysis of transcription levels of proinflammatory cytokines, type I IFN and ISGs at baseline was performed for a representative COPA^deficient THP-1 cell line (sgRNA 1). Data are mean ± SEM from 3 independent experiments and presented as fold change to the parental control cell line (THP-1 Cas9). Statistical significance was assessed by two-tailed ratio-paired Student's *t*-test. **c** Analysis of baseline levels of IFNβ and CXCL10 in cell culture supernatants by ELISA. Data are mean ± SEM pooled from 3 independent experiments, statistical analysis using two-tailed ratio-paired Student's *t*-test. **d** Immunoblotting of phosphorylated TBK1 (pTBK1) in COPA^deficient HeLa cells generated by CRISPR/Cas9 targeting of COPA with sgRNA 1. Representative result of $n = 3$ independent experiments. **e** Assessment of the baseline gene transcription profile of COPA^deficient HeLa cells by qRT-PCR analysis. Data are presented as described in **b**. Statistical analysis using two-tailed ratio-paired Student's *t*-test. P values are indicated by numbers or as $^*P < 0.05$, $^{**}P < 0.01$. Source data are provided as a Source Data file for Fig. 1a, d.

and cell surface TLR5[28], was not able to ameliorate the inflammatory phenotype in COPA^deficient THP-1 cells (Supplementary Fig. 2d).

Another candidate immune sensor is STING, which triggers inflammation downstream of multiple cytoplasmic nucleic acid sensors including cGAS[16,29,30]. Inactive STING localizes on the ER membrane and requires coatomer complex II (COPII)-mediated anterograde trafficking to translocate to Golgi compartments upon activation[31–35]. Although COPA is not a part of the COPII complex, all compartments of the secretory pathway form a tightly regulated and interdependent network within the cell[36]. Therefore, we hypothesized that reduced functionality of COPA-mediated retrograde transport might potentially interfere with STING trafficking and cause aberrant signalling. In line with this, we identified COPA as a potential interaction partner for STING in an unbiased mass spectrometry-based quantitative proteomics experiment, where we analysed protein lysates from HEK293T cells following the pulldown of overexpressed mCitrine (mCit)-tagged human STING (Fig. 2a, Supplementary Data 1). Furthermore, this screen identified SURF4 as a potential STING-interactor, which was recently shown to act as adapter protein

mediating STING-COPA interaction for subsequent retrograde trafficking[37,38]. Additionally, other known STING-interacting proteins were found to be enriched in STING pulldowns, including TBK1. Although not significantly enriched, other COPI/COPII subunit proteins and ER- and Golgi-resident proteins were identified, highlighting the importance of the secretory pathway for STING regulation (Supplementary Data 1). Interestingly, transmembrane protein 214 (TMEM214), involved in ER stress-induced caspase-4 activation and induction of apoptosis[39], was highly enriched in STING pulldowns; however, a functional link to STING has not yet been described in the literature and was not further investigated in this study.

CRISPR/Cas9-mediated genetic co-deletion of STING ameliorated the spontaneous pSTAT1 signal (Fig. 2b) as well as inflammatory gene expression in COPA^deficient THP-1 cells (Fig. 2c). Interestingly, TNF transcription levels remained high after STING deletion and are likely the result of NF-κB pathway activation following ER stress, as previously suggested[1]. We independently confirmed this result by genetic deletion of STING in COPA^deficient HeLa cells, which ameliorated type I IFN-mediated inflammatory baseline signalling and the pTBK1 signal

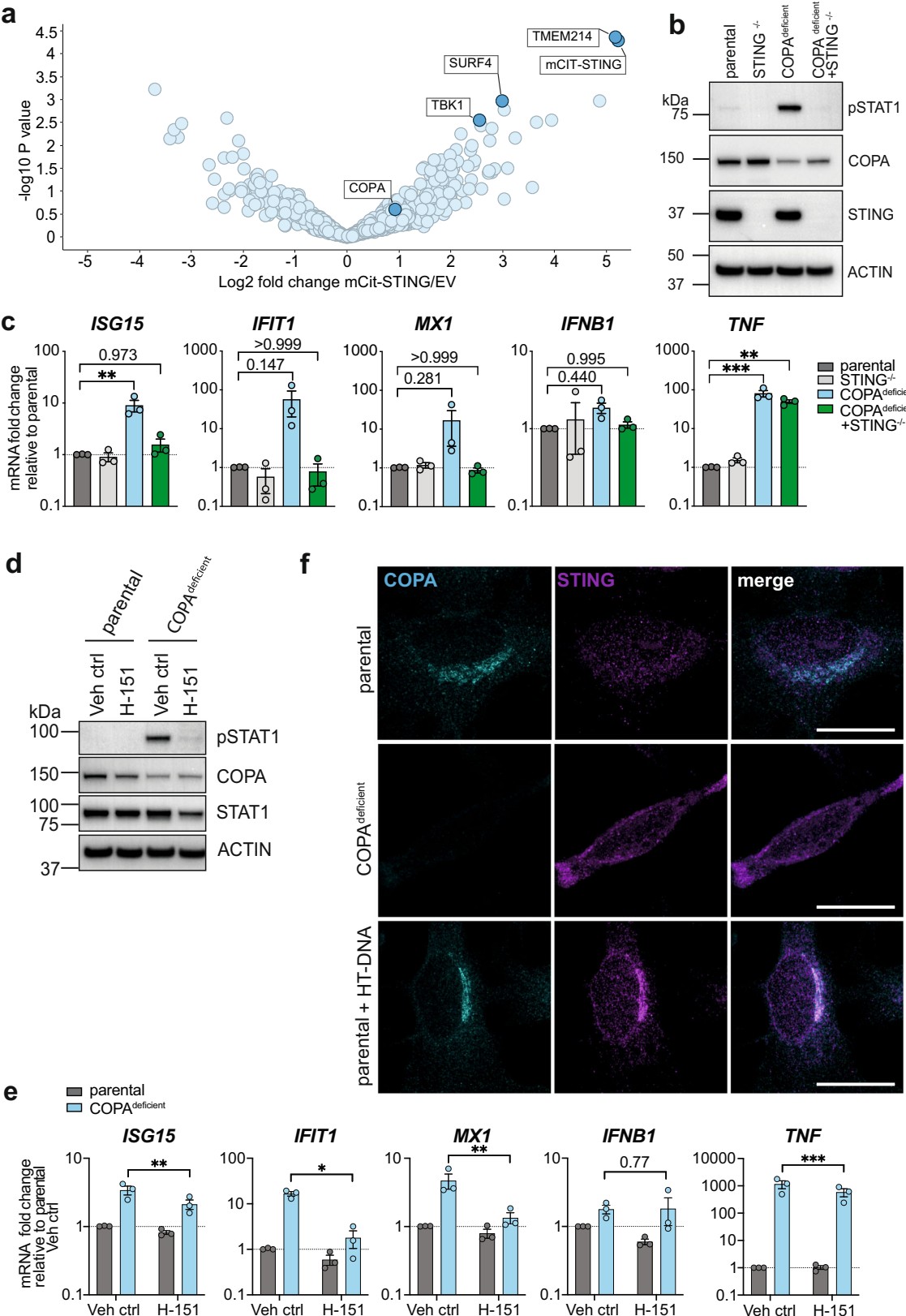

(Supplementary Fig. 3). Furthermore, in the context of potential pharmaceutical intervention, treatment of COPA^deficient THP-1 cells with small-molecule STING inhibitor H-151 was able to markedly reduce baseline STAT1 phosphorylation (Fig. 2d) and type I IFN-mediated gene transcription (Fig. 2e), thus indicating a possible targeted treatment for COPA syndrome patients.

ER-to-Golgi translocation of STING is a prerequisite for interaction with downstream signalling molecules[10,17,40]. We therefore aimed to investigate the cellular localization of STING in COPA-deficiency and performed immunofluorescence (IF) staining of endogenous STING in HeLa cells (Fig. 2f). As positive control, parental HeLa cells were stimulated with HT-DNA, a

**Fig. 2 Inflammation induced by COPA-deficiency is STING dependent. a** Volcano plot representing the $\log_2$ protein ratios of potential STING-interacting proteins identified by mass spectrometry-based quantitative proteomics after transient overexpression of mCit-STING and pulldown in HEK293T cells relative to empty vector (EV) control ($n = 3$ independent IPs). COPA may participate in a protein complex with STING. Differential expression analysis was performed using *limma* with an adjusted p-value cutoff at 0.05 and log2 fold change cutoff at 1. **b** Representative western blot of COPA[deficient] THP-1 cells or co-deleted for COPA and STING (COPA[deficient]/STING[−/−]) after 72 hrs Dox treatment. Representative result of $n = 3$ independent experiments. **c** qRT-PCR analysis of transcription levels in THP-1 cells at baseline (72 hrs Dox). Data are mean ± SEM from 3 independent experiments. Statistical significance was calculated by one-way ANOVA and Dunnett's multiple comparison test. **d** Representative western blot of COPA[deficient] THP-1 cells after treatment with STING inhibitor H-151 (2.5 μM) or vehicle control (Veh ctrl). Representative result of $n = 3$. **e** Analysis of proinflammatory gene and ISG transcription levels in COPA[deficient] THP-1 cells following treatment with H-151 (2.5 μM). Data are shown as mean ± SEM from 3 independent experiments. Statistical analysis by two-tailed ratio paired Student's *t*-test comparing control and H-151 treatment individually for each cell line. **f** Super-resolution immunofluorescences microscopy of parental and COPA[deficient] HeLa cells stained for COPA (cyan) and endogenous STING (magenta). Parental cells transfected with HT-DNA (2 μg/ml, 2 hrs) show typical puncta formation which represent STING accumulation at Golgi compartments as indication of STING activation. Representative experiment shown of $n = 3$, scale bar represents 20 μm. P values are indicated by numbers or as *$P < 0.05$, **$P < 0.01$, ***$P < 0.001$. Source data are provided as a Source Data file for Fig. 2b, d.

double-stranded DNA molecule to activate cGAS-induced STING ER-to-Golgi translocation which can be observed as distinct puncta formation (Fig. 2f). However, in COPA[deficient] HeLa cells without further stimulation, puncta formation of endogenous STING was not as clear and rather suggested the formation of multiple smaller size specks distributed throughout the cytoplasm (Fig. 2f). Since the overall signal of stained endogenous STING was quite weak, we stably overexpressed STING-GFP in these cell lines via retroviral transduction. Parental cells showed the diffuse cytoplasmic distribution of STING-GFP when untreated and the formation of puncta that co-localized with cis-Golgi marker GM130 following HT-DNA transfection (Supplementary Fig. 4a, b). In COPA-deficient cells, multiple STING-GFP speckles were observed, which largely co-localized with cis-Golgi marker (GM130) (Supplementary Fig. 4a, b). Co-localization with ER marker KDEL was not different when compared to parental cells (Supplementary Fig. 4c). Considering that loss of COPA is associated with Golgi dispersal[41] (Supplementary Fig. 4, 5) and spontaneously accumulated STING-GFP co-localized with Golgi but not ER markers, these experiments suggest that STING signalling occurs due to spontaneous accumulation at the dispersed Golgi in this in vitro model of COPA syndrome.

Overall, these results suggest that spontaneous STING activation is driving the inflammatory response in COPA syndrome model THP-1 and HeLa cell lines, which is likely the result of STING accumulation at dispersed Golgi fragments.

**Mutations in COPA drive STING-dependent inflammation**. In order to validate our findings in patient samples, we analysed PBMCs from a COPA syndrome patient carrying the c.698 G > A (R233H) mutation with clinical presentation of severe polyarticular arthritis and lung disease[6]. At the time of sample collection, the patient was treated with prednisone and rituximab. Ex vivo flow cytometry analysis revealed elevated levels of pTBK1, particularly in CD14-expressing monocytes (Supplementary Fig. 6). Treatment with STING inhibitor H-151 was able to ameliorate baseline TBK1 phosphorylation as shown in representative histograms and quantified as fold change of pTBK1 levels after H-151 treatment (measured by geometric mean fluorescence intensity (MFI)), thereby confirming basal STING activation in patient PBMCs (Fig. 3a, b).

To further study inflammation in the context of COPA syndrome, we generated overexpression plasmids encoding the previously published loss of function COPA mutations E241K and R233H[1]. Initially, we overexpressed these constructs in HEK293T cells which lack detectable levels of endogenous STING[16,42] (Supplementary Fig. 7). Phosphorylation of TBK1 and IRF3 was only observed when myc-tagged COPA was co-expressed with mCit-tagged STING (Fig. 3c). However, in this

system, STING is transiently overexpressed at high levels. This results in strong baseline activation of WT STING alone, observed by above background pTBK1 and pIRF3 signal (Fig. 3c), likely caused by spontaneous dimerization-induced Golgi translocation and subsequent pathway activation[43], which is a commonly observed disadvantage of this experimental setup. STING signalling was slightly elevated by co-expression with WT COPA. Rather than a specific effect of WT COPA, this observation is likely the result of high-level protein translation following co-transfection that overwhelms ER folding and retrograde transport capacities and subsequently causes STING retention at the Golgi that promotes signalling. Interestingly, when co-expressed with COPA mutants STING activation was markedly increased, which suggests that impaired retrograde trafficking prevents Golgi-to-ER retrieval and promotes STING accumulation at Golgi compartments, which results in increased signalling (Fig. 3c).

Therefore, we sought to test a more physiological setting with lower, stable expression of STING-GFP incorporated via lentiviral transduction of HEK293T cells and subsequent FACS sorting for low expression close to endogenous level. Surprisingly, under these conditions, the overexpression of COPA mutants E241K and R233H was not able to drive inflammation (Fig. 3d). In order to independently confirm this finding, we transiently transfected COPA E241K and R233H using 2 different plasmid concentrations in HEK293 cells, which express endogenous STING at levels comparable to HeLa cells (Supplementary Fig. 7). Again, in this system, overexpression of COPA mutants was not able to induce pIRF3 over empty vector (EV) control levels (Fig. 3e).

Importantly, these experiments suggest that inflammation driven by COPA mutations in HEK239T cells only occurs when STING is overexpressed. However, with endogenous levels of STING expression, as in HEK293 cells or HEK293T cells stably expressing STING-GFP, overexpressed COPA mutations cannot drive inflammation, likely because an essential player of the COPA syndrome pathology is lacking in these cell lines. Given that all the downstream machinery required for STING signalling appears to be intact in HEK293T cells (Fig. 3c) we wondered if the missing part of the pathway was actually upstream, specifically the cytoplasmic DNA sensor cGAS, since it is not endogenously expressed in HEK293(T) cells (Supplementary Fig. 7).

**Inflammatory signalling in COPA[deficient] THP-1 cells is cGAS-dependent**. In order to investigate the involvement of cGAS in COPA syndrome pathology, CRISPR/Cas9 gene editing was employed to delete COPA in a monoclonal cGAS knockout (cGAS[−/−]) THP-1 cell line[44]. Lack of cGAS completely abolished spontaneous phosphorylation of STAT1, which was re-

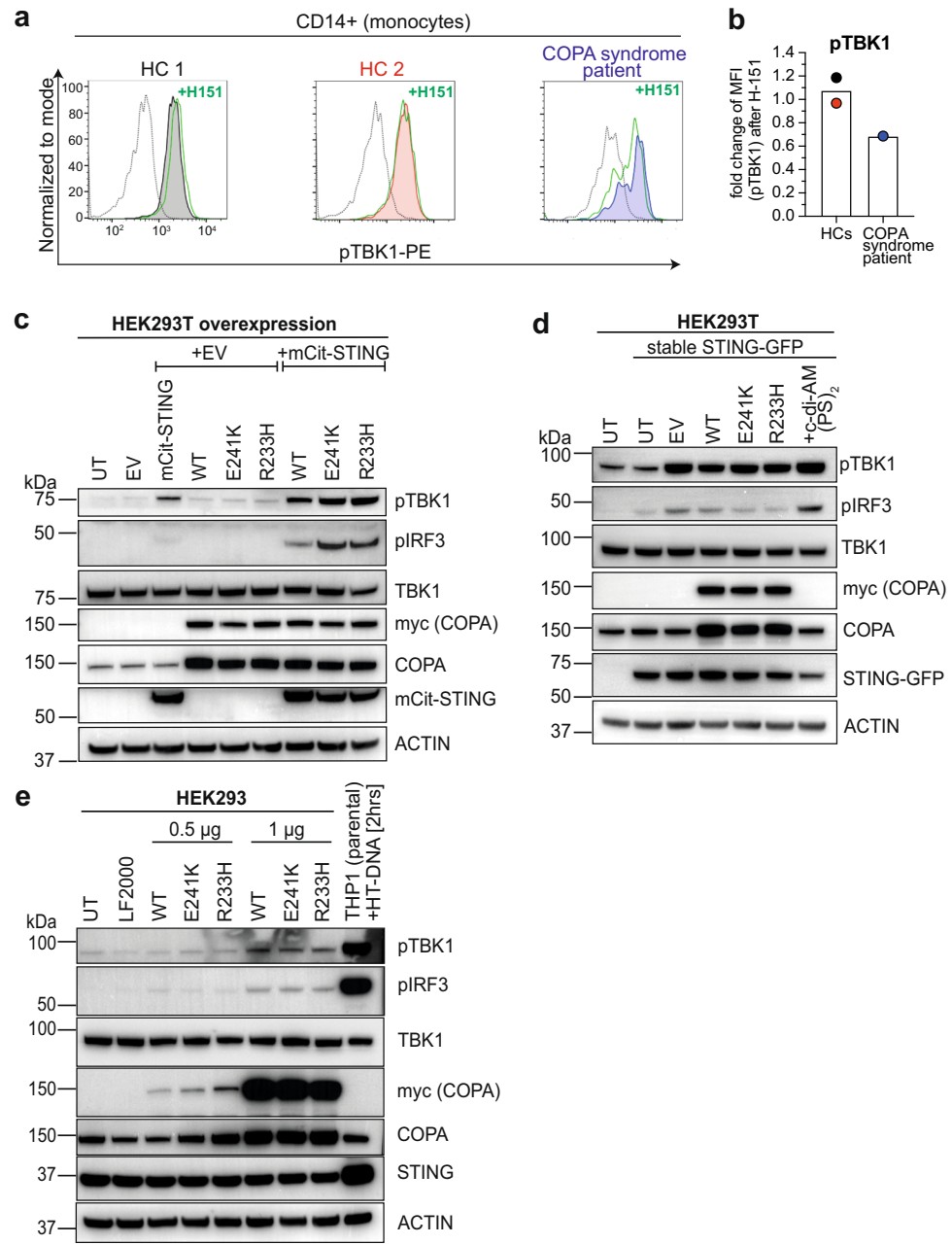

**Fig. 3 Mutations in COPA trigger STING-dependent inflammation. a** Representative histograms of flow cytometry analysis for phosphorylated TBK1 (pTBK1, used fluorochrome PE) in monocytes (CD14 + /CD3-) isolated from PBMCs of one COPA syndrome patient (blue)[6] and two healthy control individuals (HCs) (black, red) before and after treatment with STING inhibitor H-151 (5 µM) for 4 hrs (green line). Dotted line represents isotype control. $n$ = 1. **b** Quantification of data shown in (**a**). Bar graph represents fold change in pTBK1 geometric mean fluorescence intensity (MFI) following H-151 treatment relative to untreated control. Data presented as mean of two independent HCs from $n = 1$ experiments. **c** Western blot analysis of HEK293T cells following co-overexpression of mCit-STING and COPA mutants E241K and R233H (0.5 µg DNA per construct) 24 hrs after transfection. Representative result of $n = 3$. **d** Immunoblot analysis of stably STING-GFP-expressing HEK293T cells 24 hrs after transient transfection with 0.5 µg plasmid DNA encoding COPA WT or mutants. Untransfected cells stimulated with c-di-AM(PS)$_2$ (20 µM, 2 hrs) are shown as control (last lane). Representative result of $n = 3$ experiments. **e** HEK293 cells (express endogenous STING) were transiently transfected with COPA WT or mutants (0.5 µg or 1 µg DNA/well) and harvested for western blot analysis after 24 hrs. As a positive control, cell lysate of THP-1 cells stimulated with HT-DNA (2 µg/ml, 2 hrs) was used. Representative experiment of 3 independent repeats. Source data are provided as a Source Data file for Fig. 3c, d, e.

established upon reconstitution with GFP-tagged cGAS[45] via lentiviral transduction (Fig. 4a). Similarly, transcription levels of proinflammatory cytokines and ISGs were ameliorated in COPA$^{deficient}$ cells when cGAS was deleted (Fig. 4b). Interestingly, co-deleted COPA$^{deficient}$/cGAS$^{-/-}$ THP-1 cells showed stronger reduction in *TNF* gene transcription compared to STING deletion in COPA$^{deficient}$ THP-1 cells (Fig. 2c). One

reason for this could be that the cGAS$^{-/-}$ cells are monoclonal and completely lack cGAS expression[44], whereas the STING$^{-/-}$ cells are a mixed population of CRISPR/Cas9-deleted cells, with some residual STING expression. However, clonal cGAS deletion was still not able to fully reverse *TNF* transcription to baseline levels, which supports the involvement of ER stress as additional NF-κB activating pathway[1].

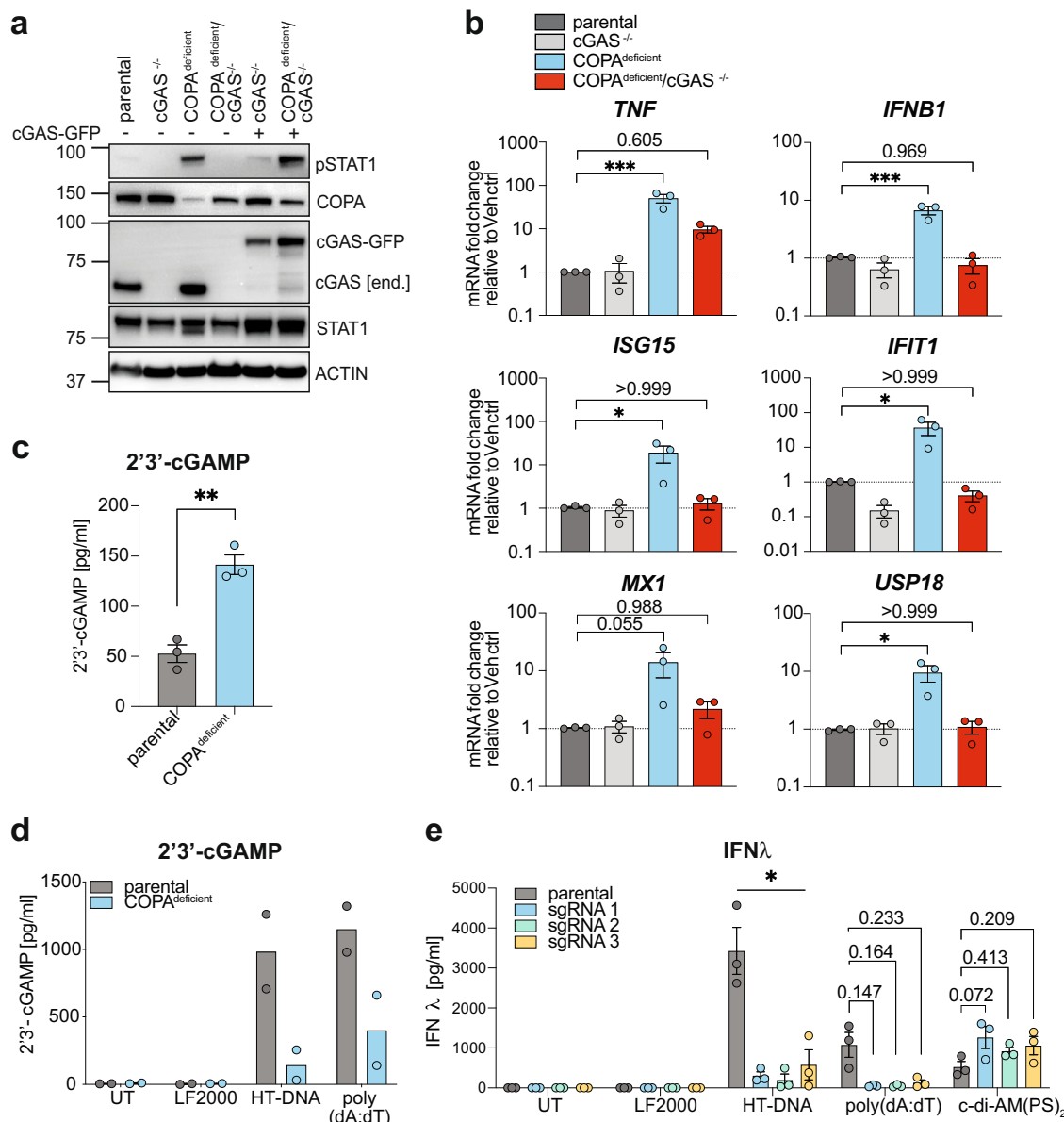

**Fig. 4 Inflammatory signalling in COPA^deficient THP-1 cells requires cGAS. a** Immunoblot analysis of parental THP-1 cells and monoclonal cGAS^−/− THP-1 cells following genetic deletion of COPA and reconstitution with cGAS-GFP via lentiviral transduction. Cells were treated with Dox for 72 hrs. A representative experiment is shown (n = 2). end.; endogenous. **b** qRT-PCR analysis of baseline ISGs and proinflammatory gene transcription profile in THP-1 cells with indicated genotypes. Data are mean ± SEM from 3 independent experiments. Statistical significance was assessed by one-way ANOVA with Dunnett's multiple comparison test. **c** Baseline analysis of 2'3'-cGAMP levels in cell lysates of parental and COPA^deficient THP-1 cells by ELISA. Data are shown as mean ± SEM from 3 independent experiments. Statistical testing by two-tailed unpaired Student's t-test. **d** COPA^deficient THP-1 cells were stimulated with cGAS activators HT-DNA (2 µg/ml) and poly (dA:dT) (1 µg/ml) for 24 hrs and 2'3'-cGAMP levels in cell lysates measured by ELISA. Data are presented as mean from n = 2 independent experiments showing individual data points. **e** 3 different COPA syndrome model THP-1 cell lines (generated and described in Fig. 1a) were stimulated with HT-DNA (2 µg/ml), poly (dA:dT) (1 µg/ml) as well as STING activator c-di-AM(PS)$_2$ (20 µM) for 24 hrs. Supernatants were analysed for released IFNλ by ELISA. Data are mean ± SEM from 3 independent experiments. Statistical significance was assessed by one-way ANOVA using parental cells as comparator group individually for each stimulus. P values are indicated by numbers or as * P < 0.05, ** P < 0.01, *** P < 0.001. Lipofectamine 2000, LF2000. Source data are provided as a Source Data file for Fig. 4a.

Recently, the role of SURF4 as an adapter protein that mediates packaging of STING into COPI vesicles was established[37,38]. In support of this, our mass spectrometry analysis identified SURF4 as an interactor of STING after overexpression in HEK293T cells (Fig. 2a). Assuming that lack of SURF4 specifically blocks Golgi-to-ER trafficking of STING and other SURF4 target proteins, we found increased phosphorylation of STAT1 and elevated transcription of ISGs following the genetic deletion of SURF4 in THP-1 WT cells (Supplementary Fig. 8; preliminary data). This

phenotype was ablated in cGAS^−/−/SURF4^−/− THP-1 cells (Supplementary Fig. 8, preliminary data) further supporting the requirement of cGAS as the initial driver for STING accumulation at the Golgi and subsequent inflammatory signalling in COPA syndrome.

To further confirm cGAS activation, we analysed cytoplasmic levels of cGAS-produced second messenger 2'3'-cGAMP by ELISA. Surprisingly, 2'3'-cGAMP levels in COPA^deficient THP-1 cells were significantly elevated over parental controls (Fig. 4c).

This result suggests cGAS activation above tonic levels in COPA-deficiency, however, whether this is the primary defect driving COPA syndrome pathology remains to be validated in COPA mutant-expressing cells.

We then sought to investigate cGAS/STING pathway activity upon cGAS stimulation with foreign DNA (HT-DNA and poly (dA:dT)) and analysed cytoplasmic 2'3'-cGAMP production. Surprisingly, after stimulation, lower 2'3'-cGAMP levels were detected in cells lacking COPA, suggesting reduced cGAS activation following targeted stimulation (Fig. 4d). In order to independently confirm this result, three different COPA-deficient THP-1 cells lines (from Fig. 1a) were stimulated with HT-DNA, poly (dA:dT) and small-molecule STING activator 2'3'-c-di-AM(PS)$_2$ (abbreviated: c-di-AM(PS)$_2$). Cellular supernatants were collected and analysed for levels of IFNλ, a type III IFN which is similarly activated to type I IFN upon PRR stimulation[46,47] (Fig. 4e). In line with the previous result, all tested COPA-deficient cell lines were less responsive to cGAS stimulation when compared to parental control cells. Yet interestingly, direct STING stimulation by the cGAMP analogue was not impaired and comparable across all cell lines (Fig. 4e).

Together, this data demonstrates that cGAS is required for inflammation driven by COPA-deficiency in THP-1 cells, and that the second messenger 2'3'-cGAMP is elevated in these cells, indicating increased cGAS activity. However, despite this increased activation, it is no longer possible to activate cGAS with transfected cytoplasmic DNA, indicating that this is a breakpoint in the pathway that is impacted in COPA syndrome.

**Spontaneous cGAS/STING signalling is activated by general deficiency in COPI-mediated retrograde transport**. Besides COPA (subunit α), 6 other subunit proteins COPB1 (β), COPB2 (β'), COPD (δ), COPE (ε), COPG (γ), COPZ (ξ) as well as the small GTPase ARF1 are essential for functional retrograde trafficking between Golgi and ER and within cis-Golgi compartments (Fig. 5a)[2]. Here, we sought to determine whether activation of the cGAS/STING signalling pathway is uniquely linked to loss of COPA function, or whether disruption of the retrograde trafficking route in general results in inflammation via this pathway. Therefore, we randomly selected COPI-subunits COPG1, COPD and COPE and used the CRISPR/Cas9 technology to delete these subunits in HeLa and THP-1 cells.

To confirm that retrograde transport is indeed impaired upon silencing of the selected COPI subunits, we used immunofluorescence super-resolution microscopy and investigated the localization and intensity of KDEL staining in COPI subunit-deficient HeLa cells (Fig. 5b). The amino acid sequence KDEL is an ER-specific retention signal encoded on the C-terminus of soluble ER-resident proteins that undergo Golgi-to-ER retrieval through COPI-mediated retrograde transport[48]. We therefore hypothesized that impaired retrograde transport would result in the accumulation of KDEL signal within Golgi compartments. Overall, COPI-deficient cells showed increased KDEL signal intensity compared to parental controls (Fig. 5b), which is likely the consequence of increased expression of KDEL-tagged ER chaperones following ER stress and UPR activation[1,49,50]. Indeed, when compared to parental cells with intact retrograde transport, a significant shift in KDEL localization towards Golgi compartments was observed in COPA$^{\text{deficient}}$, COPG1$^{\text{deficient}}$ and COPD$^{\text{deficient}}$ HeLa cells (Fig. 5b). Quantification of this result is shown as the area ratio of KDEL signal overlaying with cis-Golgi marker GM130 divided by total area of KDEL staining per cell (Fig. 5c). These results indicate that genetic deletion of COPA, COPG1 or COPD results in impaired retrograde trafficking, which is similar to the functional defect described

for COPA mutants[1]. Immunoblotting confirmed that unless intentionally deleted, only minor variability is observed for COPA, COPG1 and COPD protein expression levels across COPI subunit-deficient THP-1 cell lines (Fig. 5d). Therefore, despite deletion of selected subunits, expression levels of tested residual COPI proteins remain largely stable suggesting that destabilization or complete co-depletion of subunits does not occur. As one exception, COPE expression levels are markedly reduced in COPA$^{\text{deficient}}$ cells (Fig. 5d). In yeast, COPE was shown to be non-essential for functional retrograde transport, but its role as a structural component to stabilize COPA at higher temperatures has been described[51]. Since COPE expression levels are dependent on COPA binding and coatomer integration[52], co-depletion of both proteins in COPA$^{\text{deficient}}$ cells is not surprising, however, this was not observed when COPE was intentionally deleted. This indicates that COPE is not required for COPA stability and functional retrograde transport in human cells under the culture conditions used here. In line with this, only deficiency in COPI complex proteins COPA, COPG1 and COPD but not COPE resulted in spontaneous inflammatory signalling (Fig. 5d,e). In order to determine whether the inflammatory phenotype can be ameliorated by inhibition of cGAS/STING signalling, COPI subunit-deficient THP-1 cells were treated with STING inhibitor H-151. Immunoblot as well as qRT-PCR analysis showed reduction of STAT1 phosphorylation (Fig. 5d) and decreased transcription of proinflammatory genes and ISGs after inhibitor treatment (Fig. 5e). These data demonstrate that indeed, the cGAS/STING pathway is activated upon deletion of different COPI subunit proteins, and therefore a general defect in retrograde transport sufficiently induces inflammatory signalling. Consequently, aberrant STING signalling is unlikely to be the unique result of reduced COPA function.

Also similar to the loss of COPA, COPG1-deficiency results in a break in cGAS signalling upon stimulation with HT-DNA and poly (dA:dT), but IFNλ release following STING stimulation with c-di-AM(PS)$_2$ remains comparable to control cells or higher (Supplementary Fig. 9), further highlighting the essential role of intact retrograde transport for functional cellular defence in response to foreign DNA.

**Retrograde trafficking between Golgi and ER is the transport direction underlying STING mislocalization in COPI-deficiency**. Besides Golgi-to-ER retrieval of proteins, the COPI complex is also required for trafficking within Golgi compartments[53]. Disruption of which of these transport directions causes defective STING regulation in COPA syndrome has not yet been experimentally examined.

Within the cells, trafficking of cargo proteins via different COPI transport routes is thought to be regulated through multiple COPI complex isotypes[54], consisting of different combinations of subunits γ and ζ paralogues termed γ1 (COPG1), γ2 (COPG2), ζ1 (COPZ1), ζ2 (COPZ2)[55,56]. Localization studies revealed spatial segregation of COPG paralogues, with COPG1 and COPG2 being predominantly enriched in cis-Golgi or trans-Golgi compartments, respectively[57], which was suggested to reflect functional heterogeneity and specific roles. Based on cellular localization, COPG1 was suggested to be more likely involved in mediating retrograde transport between Golgi and ER while COPG2 is thought to serve the intra-Golgi transport route[57]. Taking advantage of this proposed functional distinction, we deleted COPG1 and COPG2 isoforms individually in THP-1 WT cells using the CRISPR/Cas9 approach. Deletion efficiency was confirmed by immunoblotting for COPG1 and COPG2 transcript analysis by qRT-PCR due to the lack of a specific antibody (Fig. 6a, b). Interestingly, only deletion of COPG1

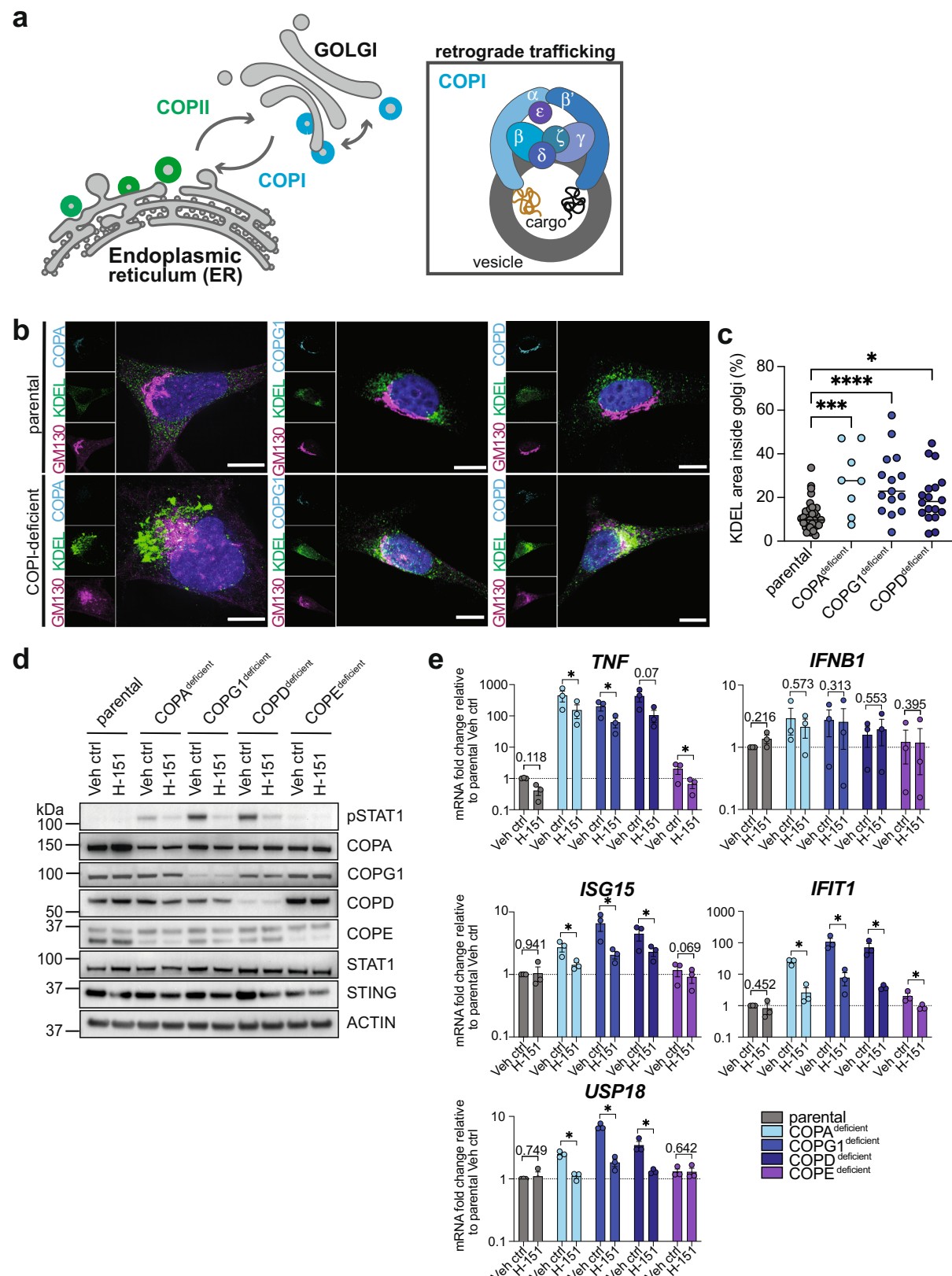

resulted in spontaneous phosphorylation of STAT1 and inflammatory gene transcription, whereas this was not observed in COPG2^deficient cells (Fig. 6a, c). Amelioration of inflammatory signalling in COPG1^deficient cells was achieved by treatment with STING inhibitor H-151 (Fig. 5d, e), indicating STING pathway activation in COPG1^deficient THP-1 cells due to defective

retrograde transport, whereas deletion of COPG2, implicated in intra-Golgi transport, had no effect.

Despite the described differences in localization of COPG1- and COPG2-containing COPI vesicles within secretory pathway compartments[57], the suggested functional difference between both paralogues has not yet been confirmed experimentally[58].

**Fig. 5 cGAS/STING pathway activation is caused by general deficiency in COPI-mediated retrograde trafficking. a** Schematic illustration of coatomer complexes COPII (green) and COPI (blue) mediating intracellular trafficking between endoplasmic reticulum (ER) and Golgi compartments. The COPII complex mediates anterograde transport from ER to Golgi, whereas COPI mediates retrograde transport from Golgi to ER as well as within cis-Golgi compartments. COPI vesicle formation occurs at the Golgi membrane through interaction of 7 subunits: α (COPA), β (COPB1), β′ (COPB2), δ (COPD), ε (COPE), γ (COPG), ζ (COPZ). After vesicle budding, the coatomer coats are shed off and released into the cytoplasm to allow vesicle fusion at the target membrane[2]. **b** Representative IF images of parental, COPA^deficient, COPG1^deficient and COPD^deficient HeLa cells co-stained for KDEL (green), GM130 (magenta), indicated COPI subunit (cyan) and DAPI (blue). Images are presented as maximum intensity projection. Scale bar 10 μm. **c** Quantification of IF co-localization of KDEL and GM130 signal in COPI subunit-deficient HeLa cells shown in (**b**). Results are quantified as percentage area of KDEL localization (ER-specific retention signal) inside cis-Golgi (GM130). Each dot represents one single cell. Parental cells stained for different subunits were combined for quantification analysis. Statistical significance was assessed by one-way ANOVA with Dunnett's multiple comparison test using $n = 35$ (parental), $n = 8$ (COPA^deficient), $n = 15$ (COPG1^deficient) and $n = 18$ (COPD^deficient) biologically independent cells examined in 2 independent experiments. Line at median. **d** Immunoblot analysis of THP-1 cells after CRISPR/Cas9-mediated genetic deletion of COPI-subunit proteins COPA, COPG1, COPD and COPE following treatment with STING inhibitor H-151 (2.5 μM). Results are shown as a representative of $n = 3$ independent repeats **e** qRT-PCR analysis of proinflammatory genes and ISGs in COPI-subunit deficient THP-1 cells following H-151 treatment (2.5 μM). Data are shown as mean ± SEM from $n = 3$ independent experiments and statistical significance was assessed by two-tailed ratio paired Student's $t$-test comparing Veh ctrl and H-151 treatment individually for each cell line. $P$ values are indicated by numbers or as * $P < 0.05$, ** $P < 0.01$, *** $P < 0.001$, **** $P < 0.0001$. Source data are provided as a Source Data file for Fig. 5c, d.

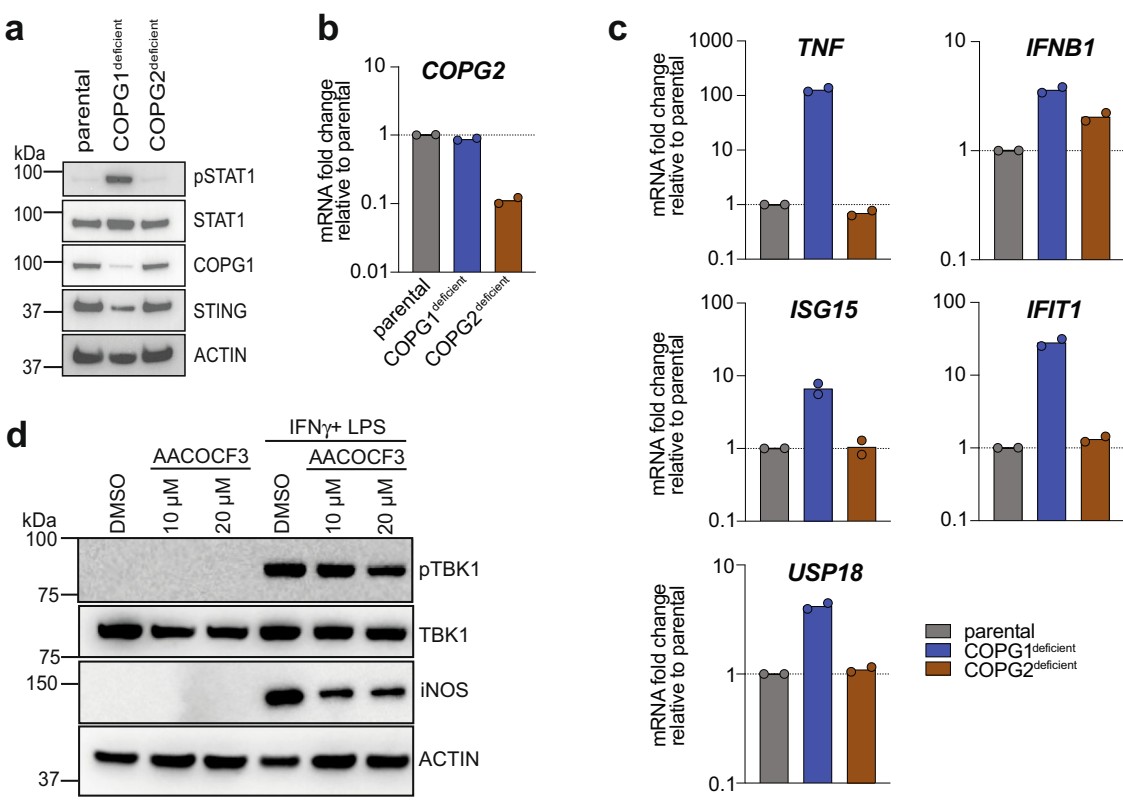

**Fig. 6 Targeted inhibition of anterograde intra-Golgi transport does not activate spontaneous STING signalling. a** Representative immunoblot analysis of baseline signalling in THP-1 cells after targeted CRISPR/Cas9-mediated deletion of COPG1 or COPG2 paralogues. $n = 2$. **b** *COPG2* transcription levels of THP-1 cells used in (**a**) analysed by qRT-PCR. Data are presented as mean from $n = 2$ independent experiments showing individual data points. **c** qRT-PCR analysis of baseline inflammatory cytokine transcripts in COPG1- and COPG2-depleted THP-1 cells. Data are presented as mean from $n = 2$ independent experiments showing individual data points. **d** Representative western blot analysis of iBMDM cells after treatment with cPLA2α inhibitor AACOCF3. Spontaneous activation of STING signalling was evaluated by immunoblotting for pTBK1, using ACTIN as a loading control. Inhibitor activity was analysed by its effect on iNOS expression levels following IFNγ priming (50 ng/ml, overnight) and LPS stimulation (25 ng/ml, 6 hrs) in absence or presence of AACOCF3 (10 and 20 μM, 30 min preincubation). A representative experiment of $n = 3$ is shown. Source data are provided as a Source Data file for Fig. 6a, d.

Proteomic profiling of γ1ζ1, γ1ζ2, γ2ζ1−COPI vesicle contents did not reveal significant differences[59], however, mediation of specific transport directions has not been investigated.

Therefore, in an alternative approach, we specifically blocked intra-Golgi trafficking by pharmacological inhibition of the cytosolic $Ca^{2+}$-dependent phospholipase $A_2$ alpha (cPLA2α)[60], which is required for the formation of intercisternal tubules that connect Golgi stacks and are essential for intra-Golgi transport[60,61]. Treatment of immortalized bone marrow-derived macrophages (iBMDMs) with cPLA2α inhibitor arachidonyl trifluoromethyl ketone (AACOCF3) did not result in spontaneous phosphorylation of TBK1, indicating that defective intra-Golgi trafficking does not sufficiently activate spontaneous STING signalling (Fig. 6d). Since cPLA2α was shown to be involved in

regulating inducible nitric oxide synthase (iNOS) expression[62], we confirmed AACOCF3 activity by evaluating expression levels of iNOS in response to IFNγ and lipopolysaccharide (LPS) stimulation. Pretreatment with cPLA2α inhibitor reduced iNOS levels (Fig. 6d), thereby validating inhibitor functionality at the used concentrations[63], although off-target effects cannot be ruled out.

COPI-mediated formation of early endosome has been shown to be required for autophagy[41]. Importantly, functional autophagy and lysosomal degradation pathways are required for the termination of STING signalling[34,64,65]. Therefore, a possible defect in those pathways caused by COPA deficiency was excluded by monitoring STING expression levels during Dox-induced COPA depletion (96 hours (hrs) Dox treatment) in THP-1 cells (Supplementary Fig. 10). Progressive loss of COPA did not coincide with the accumulation of STING, suggesting the normal function of STING degradation pathways.

Together these findings indicate, that although COPA as part of the COPI complex mediates multiple transport directions within the secretory pathway, the disease-causing mechanism in COPA deficiency relies on defective retrograde trafficking, specifically lost Golgi-to-ER retrieval of immune signalling protein STING.

Collectively, our results demonstrate that aberrant STING signalling in cell lines lacking COPA is the result of defective retrograde trafficking which impairs steady-state Golgi-to-ER retrieval of STING and causes subsequent accumulation at Golgi compartments. Spontaneous inflammation was dependent on upstream dsDNA detector cGAS in THP-1 cell lines, which suggests the requirement of a STING-activating stimulus to induce pathogenicity. However interestingly, cGAS no longer responded sufficiently to cytoplasmic DNA stimulation suggesting a break in the cGAS/STING signalling axis due to COPA deficiency. The finding that deletion of other COPI subunits, such as COPG1 or COPD, impart a similar cellular phenotype suggests that immune-mediated disorders associated with mutations in other COPI subunit proteins may exist.

## Discussion
In this study, we have generated model cell lines that recapitulate the type I IFN signature observed in PBMCs of COPA syndrome patients[6]. We show that the type I IFN signature can be ablated by genetic deletion and pharmacological inhibition of STING and also its upstream regulator cGAS, a cytosolic DNA sensor. Interestingly, the clinical phenotype of COPA syndrome partially overlaps with symptoms reported in STING-associated vasculopathy with onset in infancy (SAVI), including severe systemic inflammation, recurrent fevers, interstitial lung disease, early onset in life and a predominant constitutive type I IFN gene activation[66,67]. For SAVI, multiple case reports identified gain of function mutations in STING, causing spontaneous dimerization, Golgi translocation and STING pathway activation[68]. Localization studies of endogenous STING and reconstituted STING-GFP in COPA$^{deficient}$ HeLa cells showed STING accumulation in multiple small specks scattered throughout the cytosol, which partially co-localized with dispersed Golgi fragments in preliminary experiments. Partial Golgi dispersal and redistribution has previously been observed in COPA-deficient cell lines[41] and also occurs after secretory pathway inhibition by treatment with Brefeldin-A, a small GTPase inhibitor that prevents initiation of COPI and II vesicle formation[69]. STING signalling in COPA-depleted HeLa cells therefore may occur from smaller Golgi fragments, rather than an intact Golgi complex. In line with our data, spontaneous Golgi accumulation of STING in cell lines overexpressing COPA mutants and COPA syndrome patient cells

has recently been shown in other studies published while our manuscript was under review[37,38,67]. Since Golgi dispersal was not observed in the complementary studies[9,37,38,67], this is likely a distinct feature of the COPA-deletion model system. Importantly, ligand-stimulated STING activation is retained in our COPA$^{deficient}$ cells, suggesting that the observed Golgi fragmentation does not impair STING signalling[70].

In contrast to SAVI, we have demonstrated that aberrant type I IFN signalling in THP-1 cells lacking COPA and COPA-STING-adapter protein SURF4 requires upstream activation of cGAS, which normally protects cells from infections by sensing cytoplasmic DNA. This is in line with a previous study, which proposed tonic cGAS activation by low levels of cytoplasmic mitochondrial or genomic DNA as activators of STING translocation during the steady-state[67]. Our data indicate that retrograde trafficking by COPI retrieves STING from the Golgi to prevent signalling as a key mechanism of negative regulation during homeostatic conditions. Following activation, STING binding to SURF4 and COPA is reduced[38], which results in accumulation at the Golgi and allows for interaction with downstream signalling molecules. Since the loss of function mutations in COPA reduce the COPI binding efficiency, steady-state Golgi-to-ER retrieval of the SURF4-STING complex is impaired and lowers the threshold for STING accumulation and signalling[9,37,38,67]. This proposed mechanism is supported by the relatively high incomplete penetrance of clinical symptoms (around 25 %[4]) in individuals carrying COPA mutations, indicating the requirement of additional triggers or environmental factors to disrupt the balance of homeostatic STING circulation and induce disease onset. Although not yet identified, contributing co-factors could be infections or additional predisposing mutations in genes regulating ER stress pathways or other molecules involved in inflammatory signalling via cGAS/STING.

Furthermore, we observed increased levels of 2'3'-cGAMP in COPA$^{deficient}$ THP-1 cells indicative of cGAS activation, which could be a secondary consequence of altered cellular homeostasis in COPA-deficiency. Further validation of these findings in model systems expressing COPA mutants is required, to exclude the possibility of Cas9-mediated cGAS activation in the here used experimental system[71]. cGAS activation as the primary consequence of COPA-deficiency seems less likely, since overexpression of COPA mutants in mouse embryonic fibroblasts and HEK293T cells induced aberrant STING signalling independent of cGAS[37,38,67], which demonstrates that the main pathway defect is caused by STING accumulation at the Golgi. Importantly, an activating stimulus appears to be required and provided either by overexpression of STING or baseline cGAS signalling[43,67]. However, species-specific differences in the case of murine cell lines cannot be excluded[37,38,67].

Nevertheless, secondary activation of cGAS could further aggravate inflammation, since overactivation of cGAS by self-DNA has now been implicated in several inflammatory autoimmune pathologies. For example, Aicardi-Goutières syndrome and familial chilblain lupus are caused by mutations in three-prime repair exonuclease 1 (TREX1), ribonuclease H2 (RNaseH2) or sterile alpha motif domain and HD domain-containing protein 1(SAMHD1) that result in accumulation of genomic self-DNA in the cytoplasm via different mechanisms and therefore trigger cGAS activation and autoinflammation (reviewed in[72]).

Another possibility is that cGAS activation could be the result of leakage of mitochondrial DNA into the cytoplasm[15,73,74]. Within cells, ER and mitochondria form a structural and functional interactive network[75] and both organelles exchange metabolites, signalling molecules and ions through so called mitochondrial-associated membranes[76,77]. COPA-deficiency-induced alterations in ER homeostasis may therefore impair

mitochondrial integrity and lead to release of mitochondrial DNA which could subsequently activate cGAS[78,79]. Intriguingly, cGAS has also been shown to undergo phase separation, which is described as droplet-like compartment formation for subsequent signalling[80]. It is believed that this phenomenon is highly dependent on the availability of ions, such as Zinc ($Zn^{2+}$) which are mainly stored in mitochondria and the ER[81]. Therefore, it is conceivable that increased ER stress activated by COPI-deficiency may result in uncontrolled release of $Zn^{2+}$ ions into the cytoplasm, promoting the phase separation of cGAS and subsequent signalling[80]. However, the mechanism of cGAS activation in COPA syndrome remains speculative and requires further investigation.

In our study, we show that the cGAS/STING pathway is activated upon deletion of different COPI-subunit proteins COPA, COPG1 or COPD, whereas deficiency of COPE does not induce inflammatory signalling. Therefore, one could speculate that there may be other interferonopathies associated with defective retrograde transport caused by lost function of COPI subunits other than COPA, that have not yet been described. Interestingly, loss of function mutations in *COPB2* (COPβ') or COPD (COPδ, encoded by *ARCN1* gene) are linked to diseases associated with skeletal developmental defects and microcephalus formation[82,83]. Given the manifestation of these conditions, an underlying IFNαβ signature may be present but undocumented. However similar to COPA syndrome, numerous other factors could be involved to determine whether a particular COPI deficiency results in a more pronounced developmental or inflammatory pathology.

A recent study reported a primary immunodeficiency disease associated with a novel homozygous loss of function mutation in *COPG1* (K652E) and identified impaired retrograde trafficking of KDEL receptor-bound proteins and subsequently increased ER stress in activated B and T cells as the underlying disease-causing mechanism. Treatment with the chemical chaperon tauroursodeoxycholic acid (TUDCA), an ER stress relieving agent, was able to reverse the B and T cell dysfunction in COPG1[K652E] mice upon exposure to pet store mice, however, IL-6 serum levels remained elevated[84], which suggests additional activation of an ER stress-independent inflammatory pathway. In our in vitro *model*, CRISPR/Cas9-mediated deletion of COPG1 resulted in spontaneous STING signalling. However, since levels of type I IFN and ISGs were not measured in COPG1 (K652E) patients and the clinical presentation does not overlap with symptoms of COPA syndrome, no conclusion about contribution of a STING-driven pathology can be drawn.

Interestingly, we demonstrate that deficiency of COPA and COPG1 subunits results in an inability to activate cGAS following stimulation with synthetic DNA ligands. This suggests defective innate immune defence mechanisms upon lost function of COPA and COPG1, which could perhaps contribute to immunodeficiency of COPG1 (K652E) patients. Although profound immunodeficiency has not been reported in COPA syndrome patients, recurrent respiratory and opportunistic infections were reported in individual cases[1,9,85,86]. Since immune deficiency due to loss of cGAS/STING signalling is not described in the literature, it is not clear what pathology, if any, we should expect to see associated with COPI-mediated disease.

Overall, this study shows that the type I IFN signature described in COPA syndrome patients is caused by a spontaneous activation of the cGAS/STING pathway. This finding is particularly exciting in the context of pharmacological intervention. Originally, COPA syndrome patients have been treated with various combinations of immunosuppressive drugs including DMARDs (disease-modifying anti-rheumatic drugs), NSAIDs (nonsteroidal anti-inflammatory drugs) and biologics, which were only able to partially control the disease[4]. Recently published case reports show treatment of COPA syndrome patients with JAK1/2 inhibitors ruxolitinib, baricitinib and upadacitinib[7–9]. Although the IFNαβ signature and joint inflammation were reduced successfully, patient lung function was only partially improved[7–9]. Therefore, our study provides hope that targeting the cGAS/STING pathway directly, for example with a STING inhibitor such as H-151, may be more specific and thus proves a greater beneficial approach to control type I IFN-mediated symptoms in COPA syndrome.

## Methods

**Cell culture.** Human embryonic kidney (HEK) 293 (CellBank Australia, code 85120602, (a gift from Stephen Mieruszynski, The Walter and Eliza Hall Institute of Medical Research (WEHI)), HEK293T cells (ATCC Cat. No. CRL-3216), HeLa cells (Curie Institute, Paris, a gift from Paul Gleeson, Department of Biochemistry and Molecular Biology, University of Melbourne) and immortalized bone marrow-derived macrophages (iBMDM, a gift from Daniel Simpson, WEHI, generated as described in[87]) were cultured in DMEM (Gibco, Life Technologies) supplemented with 10 % Fetal Bovine Serum (FBS, Sigma-Aldrich) and 100 U/ml Penicillin/100 μg/ml Streptomycin (Sigma-Aldrich) in a humidified incubator at 37 °C and 10 % $CO_2$. Human monocytic THP-1 cells (ATCC, Cat. No. TIB-202) were maintained in RPMI-1640 (made inhouse, RPMI 1640 powder (Life Technologies), 23.8 mM Sodium Bicarbonate ($NaHCO_3$) (Merck), 1 mM Sodium Pyruvate ($C_3H_3NaO_3$) (Sigma-Aldrich), 100 U/ml Penicillin/100 μg/ml Streptomycin) supplemented with 10 % FBS and incubated in humidified atmosphere at 37 °C and 5 % $CO_2$. The clonal cGAS KO (cGAS$^{-/-}$) THP-1 cell line was generated by Mankan and colleagues as previously published[44].

**CRISPR/Cas9 gene editing.** COPA-, COPG1-, COPG2-, COPD-, COPE-deficient and MAVS$^{-/-}$, PKR$^{-/-}$, UNC93B1$^{-/-}$, NLRP3$^{-/-}$, STING$^{-/-}$, SURF4$^{-/-}$ cells were generated using CRISPR/Cas9 gene editing as previously described[88]. Cas9 was stably expressed using the FU-Cas9-mCherry plasmid (kind gift from M. Herold, Addgene plasmid #70182)[89] or lentiCRISPR v2 Cas9 (kind gift from F. Zhang, Addgene plasmid # 52961)[90,91] in THP-1 or HeLa cells, respectively. Single guide (sg) RNAs (sequences listed in Supplementary Table 1) were cloned into the Doxycycline (Dox)-inducible FgH1t-UTG or FgH1t-UTC construct (kind gift from M. Herold, Addgene plasmids #70183, #85551)[89]. Gene deletion was induced by treatment with Dox (1 μg/ml, Sigma-Aldrich) for 72 hours (hrs), followed by 24 hrs resting period in some experiments. Protein or gene deletion was assessed by western blotting or quantitative Real-Time PCR (qRT-PCR). Stably Cas9-expressing THP-1 or HeLa cells were used as control cells and are referred to as parental cell line throughout this study.

**Stable cell line generation.** Stable STING-GFP-expressing HEK293T cells were generated via retroviral reconstitution following the method published previously[92]. Briefly, retrovirus-containing supernatant was generated by transient co-transfection of HEK293T cells (10 cm culture dish) with pMRX-hSTING-GFP (6 μg, kind gift from Nan Yan, Department of Immunology and Department of Microbiology, University of Texas), VSVg (4 μg) and gagpol (6 μg). Filtered (0.45 μm) supernatant was then used to infect HEK293T WT cells in presence of polybrene (8 μg/ml, Sigma-Aldrich), which were subsequently sorted for the low GFP-expressing population.

The reconstitution of cGAS$^{-/-}$/Cas9 and cGAS$^{-/-}$/COPA$^{deficient}$ cell lines with pTRIP-CMV-GFP-FLAG-cGAS (kind gift from N. Manel, Addgene plasmid # 86675)[45] was performed via lentiviral transduction as previously described[88]. Therefore, $1x10^6$ cells were incubated with 1 ml virus-containing supernatant and polybrene (8 μg/ml) and centrifuged for 3 hrs at 1055x *g* at 32 °C. Following 24 hrs incubation, virus-containing media was removed and STING inhibitor H-151 (2.5 μM, Life Chemicals) added in order to minimise cell death. After additional 24 hrs incubation, cells were subjected to FACS sorting for the GFP-low expressing population and grown to confluence.

**QuikChange mutagenesis.** The QuikChange Lightning Site-Directed Mutagenesis Kit (Cat. No. 210513, Agilent Technologies) was used to generate plasmids encoding the COPA syndrome patient mutations c.721 G > A (p.E241K) and c.698 G > A (p.R233H)[1] according to the manufacturer's recommendations. As template, the pCMV6Entry-COPA-myc-DDK plasmid (Origene, Cat. No. RC214199, kind gift from A. Shum, Department of Medicine, University of California San Francisco) was used. Primer sequences are listed in Supplementary Table 2.

**Cell transfection.** For overexpression studies, HEK293 or HEK293T cells were initially seeded at $2.5x10^5$ or $2x10^5$ cells/well in a 6-well plate format, respectively and transiently transfected with 0.5-1 μg of total DNA as stated in the figure or figure legend. Transfection of all plasmids including pCMV6Entry-COPA-myc-DDK, pTRIP-CMV-GFP-FLAG-cGAS, pEF-BOS-mCitrine-STING and pEF-BOS-

FLAG as an empty vector (EV) control (kindly provided by V. Hornung, LMU Munich), was performed with Lipofectamine 2000 (Life Technologies) reagent following the manufacturer's instructions. After 24 hrs incubation, cells were lysed for immunoblot analysis.

**Immunoprecipitation of mCitrine-STING for mass spectrometry-based quantitative proteomics**. Immunoprecipitation (IP) experiments were performed similarly as previously described[93]. Briefly, 3x10^6 HEK293T cells were seeded in 10 cm culture dishes. The following morning cells were transiently transfected with 10 μg of either pEF-BOS EV control or pEF-BOS-mCitrine-hSTING plasmid DNA. After 48 hrs cells were lysed in 750 μl 1xNonidet P-40 [1% NP-40, 10% glycerol, 20 mM Tris-HCl pH 7.4, 150 mM NaCl, 1 mM ethylene glycol tetraacetic acid (EGTA), 10 mM sodium pyrophosphate (NaPPi), 5 mM sodium fluoride (NaF), 1 mM sodium orthovanadate (Na₃VO₄)] supplemented with 1 mM PMSF and 1 x cOmplete^TM Protease Inhibitor Cocktail (Cat. No. 11697498001, Roche Biochemicals, Mannheim, Germany). Whole cell lysates were clarified by centrifugation at 17,000x g for 10 minutes (min) at 4 °C. For each IP sample, 1 μg of mouse IgG2A monoclonal GFP antibody (Thermo Fisher Scientific, [E36] A-11120) was crosslinked to 50 μl of Protein A Dynabeads® (Thermo Fisher Scientific, 10002D) using 5 mM BS³ (Thermo Fisher Scientific) and incubated for 30 min at RT. The crosslinking reaction was quenched by adding 1 M Tris-HCl [pH 7.4]. Antibody-crosslinked Protein A Dynabeads® were thoroughly washed before 50 μl was then added to each IP sample and incubated on a rotator at 4 °C for 2 hrs. Beads were then washed four times with 1xNP-40 buffer using a DynaMag-2 magnetic holder (Thermo Fisher Scientific) before proteins were eluted with 0.5% SDS in PBS. Eluted protein material was subjected to tryptic digestion using the FASP method as previously described[94–96]. Peptides were lyophilised using CentriVap (Labconco) prior to reconstituting in 60 μl 0.1% formic acid/2% acetonitrile. Peptide mixtures (2 μl) were separated by reverse-phase chromatography on a C18 fused silica column (I.D. 75 μm, O.D. 360 μm x 25 cm length) packed into an emitter tip (IonOpticks), using a nano-flow HPLC (M-class, Waters). The HPLC was coupled to an Impact II UHR-QqTOF mass spectrometer (Bruker) using a CaptiveSpray source and nanoBooster at 0.20 Bar using acetonitrile. Peptides were loaded directly onto the column at a constant flow rate of 400 nl/min with 0.1 % formic acid in MilliQ water and eluted with a 90 min linear gradient from 2 to 34% buffer B (99.9% acetonitrile and 0.1% formic acid). Mass spectra were acquired in a data-dependent manner including an automatic switch between MS and MS/MS scans using a 1.5 s duty cycle and 4 Hz MS1 spectra rate, followed by MS/MS scans at 8–20 Hz dependent on precursor intensity for the remainder of the cycle. MS spectra were acquired between a mass range of 200–2000 m/z. Peptide fragmentation was performed using collision-induced dissociation. Raw files consisting of high-resolution MS/MS spectra were processed with MaxQuant (version 1.6.6.0) for feature detection and protein identification using the Andromeda search engine[97] as previously described[98]. Extracted peak lists were searched against the reviewed *Homo sapiens* (UniProt, March 2019) database as well as a separate reverse decoy database to empirically assess the false discovery rate (FDR) using strict trypsin specificity, allowing up to 2 missed cleavages. LFQ quantification was selected, with a minimum ratio count of 2. PSM and protein identifications were filtered using a target-decoy approach at an FDR of 1%. Only unique and razor peptides were considered for quantification with intensity values present in at least 2 out of 3 replicates per group. Statistical analyses were performed using LFQAnalyst[99] (https://bioinformatics.erc.monash.edu/apps/LFQ-Analyst/) whereby the LFQ intensity values were used for protein quantification. Missing values were replaced by values drawn from a normal distribution of 1.8 standard deviations and a width of 0.3 for each sample (Perseus-type). Protein-wise linear models combined with empirical Bayes statistics were used for differential expression analysis using Bioconductor package *limma* whereby the adjusted P value cutoff was set at 0.05 and log2 fold change cutoff set at 1. The Benjamini-Hochberg (BH) method of FDR correction was used. The mass spectrometry proteomics data have been deposited to the ProteomeXchange Consortium via the PRIDE (https://www.ebi.ac.uk/pride/archive/projects/PXD023135)[100] partner repository with the dataset identifier PXD023135. Localization of identified STING-interacting proteins to ER or Golgi compartments in Supplementary Data 1 was determined using the Gene Ontology database (https://www.ebi.ac.uk/QuickGO/, particularly cellular component terms).

**Cell stimulation and inhibitors**. THP-1 cells were stimulated with activators of the cGAS/STING pathway, such as HT-DNA (DNA sodium salt from herring testes (Sigma-Aldrich), 2 μg/ml, transfected), poly (dA:dT) (1 μg/ml, transfected, InvivoGen), 2'3'-c-di-AM(PS)₂ (Rp,Rp) (20 μM, InvivoGen). Transfections were performed using Lipofectamine 2000 (Life Technologies) and OptiMEM (Invitrogen) according to the manufacturer's instructions.

STING inhibitor studies were performed during Dox treatment. Therefore, THP-1 cells were seeded at 8x10^4 cells/6-well plate in 2 ml complete RPMI (+ Dox 1 μg/ml). After 34 hrs incubation, STING inhibitor H-151 (2.5 μM, Life Chemicals) or DMSO (Vehicle (Veh) control) was added and incubated for 11 hrs before the inhibitor was washed off and incubation with Dox continued until the 72 hr time point was reached when cells were harvested for analysis.

To assess induction of spontaneous signalling following cPLA2α inhibition in iBMDMs, 2x10^5 cells were seeded in 400 μl complete DMEM (24-well plate format)

and incubated with AACOCF3 (10 or 20 μM, Sapphire Bioscience) or DMSO for 6 hrs. To test inhibitor functionality, iNOS expression levels were investigated after overnight priming with IFNγ (50 ng/ml, 485-MI R&D Systems), followed by 30 min incubation with AACOCF3 (10 or 20 μM, Sapphire Bioscience) or DMSO and subsequent stimulation with LPS (25 ng/ml, LPS-SM Ultrapure, # tlrl-smlps, InvivoGen) for 6 hrs.

**Immunoblotting**. 0.5–1x10^6 cells were lysed in RIPA buffer (20 mM Tris-HCl (pH 7.3), 150 mM NaCl, 5 mM EDTA, 1% Triton X-100, 0.5% sodium deoxycholate, 0.1% SDS, 10% glycerol) supplemented with cOmplete^TM Protease Inhibitor Cocktail (Cat. No. 11697498001, Roche Biochemicals), 1 mM PMSF and phosphatase inhibitors (5 mM NaF, 10 mM NaPPi, 1 mM Na₃VO₄). Using Pierce centrifuge columns (Thermo Fisher Scientific), protein lysates were purified of DNA and subsequently mixed with SDS-PAGE sample buffer. Size-based protein separation was achieved by use of 4–12% SDS PAGE gels (Novex) and NuPAGE^TM MES SDS running buffer (Thermo Fisher Scientific). Proteins were then transferred onto a PVDF membrane (Immobilon®-P Transfer Membrane, Millipore) via wet transfer and blocked in 5% skim milk/Tris-buffered saline (TBST) for 1 hr at room temperature (RT). Membranes were probed with primary antibody diluted in 5% skim milk/TBST or 5% bovine serum albumin (BSA)/TBST overnight at 4 °C: anti-COPA (Santa Cruz Biotechnology, clone H-3, sc-398099, dilution 1:1000), anti-COPD (Santa Cruz Biotechnology, clone E-12, sc-515549, dilution 1:1000), anti-COPE (Santa Cruz Biotechnology, clone A-4, sc-133195, dilution 1:1000), anti-phospho-STAT1 Tyr701 (Cell Signaling Technology, clone 58D6, #9167, dilution 1:1000), anti-phospho-TBK1/NAK Ser172 (Cell Signaling Technology, clone D52C2, #5483, dilution 1:1000), anti-phospho-IRF3 Ser386 (Abcam, ab76493, EPR2346, dilution 1:500), anti-STING (Cell Signaling Technology, clone D2P2F, #13647, dilution 1:1000), anti-STAT1 (Cell Signaling Technology, clone D1K9Y, #14994, dilution 1:1000), anti-TBK1/NAK (Cell Signaling Technology, #3013, dilution 1:1000), anti-cGAS (D1D3G, Cell Signaling Technology #15102, dilution 1:1000), anti-GFP (Life Technologies, #A11122, dilution 1:1000), anti-Myc-Tag (Cell Signaling Technology, clone 9B11, #2276, dilution 1:1000), anti-COPG (A-10, sc-393977, Santa Cruz Biotechnology, dilution 1:1000), anti-iNOS/NOS type II (BD Transduction Laboratories, #610329, dilution 1:1000), anti-NLRP3 (AdipoGen Life Sciences, Cryo-2, AG-20B-0014-C100, dilution 1:1000), anti-PKR (Santa Cruz Biotechnology, clone B-10, sc-6282, dilution 1:1000), anti-Cardif (AdipoGen Life Sciences, Adri-1, AG-20B-0004-C100, dilution 1:500) or anti-Actin-HRP (Santa Cruz Biotechnology, clone C4, sc-47778, dilution 1:10000).

Secondary antibody was incubated for 1 hr at RT (1:10,000, sheep anti-mouse IgG-HRP (Cat. No. NA931) or donkey anti-rabbit IgG-HRP (Cat. No. NA934), GE Healthcare). For development, Immobilon Forte Western HRP Substrate (Cat. No. WBLUF0500, Millipore) and the ChemiDoc Imaging touch system (BioRad) were used.

**RNA isolation and quantitative real-time PCR**. RNA was isolated using the ISOLATE II RNA Mini Kit (Cat. No. 52073, Bioline) following the manufacturer's guidelines. The Superscript III Reverse transcriptase (Cat. No. 18080-085, Invitrogen) and oligo (dT) nucleotides (Cat. No. C110B-C, Promega) were used to reverse transcribe 1 μg of total RNA. Quantitative Real-Time PCR (qRT-PCR) was performed using Maxima SYBR Green/ROX qPCR Master Mix (Cat. No. K0223, Thermo Fisher Scientific) and the ViiA 7 Real-time PCR system (Thermo Fisher Scientific). qRT-PCR Primer sequences used in this study are listed in Supplementary Table 3. Samples were run in duplicates, normalized to housekeeper gene *ACTIN* and analysed using the ΔΔCt method. Data are presented as fold change relative to vehicle control or parental cell line as indicated in the figure.

**ELISA**. To measure the production of CXCL10 and IFNβ, THP-1 cells were seeded at 1.5x10^5 cells per well into a 96-well plate after 48 hrs of Dox treatment. 24 hrs later, culture supernatants were collected and cytokine concentrations measured using the Human CXCL10/IP-10 Quantikine ELISA Kit (Cat. No. DIP100, R&D Systems) and VeriKine-HS Human IFN Beta Serum ELISA Kit (Cat. No. 41415, PBL Bioscience) following the manufacturer's instructions.

The release of IFN-λ was measured following stimulation of the cGAS/STING pathway. Cells were cultured for 72 hrs in presence of Dox before being reseeded at 0.5x10^5 cells/well into a 96-well plate format and stimulated with cGAS/STING pathway activators as described above. After 24 hrs incubation, cellular supernatants were collected and IFN-λ concentration measured using the IL-29/IL-28B (IFN-lamda 1/3) DuoSet ELISA (Cat. No. DY1598B, R&D Systems) according to the manufacturer's instructions.

To measure 2'3'-cGAMP concentration at baseline or following cGAS activation, 1x10^6 or 0.5x10^6 cells were lysed in 50 μl Thermo M-PER mammalian protein extraction reagent (Cat. No. 78503, Thermo Fisher Scientific), respectively. Lysis was performed for 10 min at 4 °C with subsequent centrifugation for 15 min at maximum speed and 4 °C. Clarified lysates were analysed using the 2'3'-cGAMP ELISA Kit (Cat. No. 501700, Cayman Chemical) following the manufacturer's instructions.

**Immunofluorescence microscopy**. Glass coverslips (18 mm x 18 mm, thickness 1½, Zeiss) were sterilised in 80% ethanol and placed in 6-well plates. After 72 hrs of

Dox treatment, HeLa cells were seeded at 3x10⁵ cells/well in 2 ml complete growth medium and left to attach for 4–6 hrs. Following treatment of indicated wells with HT-DNA (2 µg/ml, transfected) for 2 hrs, cells were washed in 1xPBS and fixed in 2 ml ice-cold methanol for 15 min at $-20\,°C$ when stained for endogenous STING. Staining for ER (KDEL) and Golgi (GM130) markers was performed after fixation with 4% paraformaldehyde (Electron Microscopy Sciences) for 15 min at RT. The cells were thoroughly washed 3x in 1xPBS and blocked (5% normal goat serum, 0.3% Triton X-100, 1xPBS) for 1 hr at RT. Primary antibodies were diluted in antibody dilution buffer (1% BSA, 0.3% Triton X-100, 1xPBS) and incubated overnight at 4 °C in a wet chamber using the following antibodies and dilutions: anti-COPA (1:100, Santa Cruz Biotechnology, clone H-3, sc-398099), anti-COPG (1:100, Cat. No. 12393-I-AP, ProteinTech), anti-COPD (1:100, Cat. No. GTX630562, clone GT1318, GeneTex), anti-GM130-AF647 (1:500, ab195303, EP892Y, Abcam), anti-STING (1:100, Cell Signaling Technology, clone D2P2F, #13647), anti-KDEL (1:200, clone 10C3, ab12223, Abcam, (Supplementary Fig. 4, 5) and anti-KDEL-AF568 (1:500, ab203421, EPR12668, Abcam, (Fig. 2f, Fig. 5b)). After 3 washes with 1xPBS, secondary antibodies (goat-anti-rabbit-AF647, Cat. No. A-21245, goat- anti-mouse-AF488, Cat. No. A-11001 (Invitrogen) were diluted (1:1000) in antibody diluent and incubated for 1 hr at RT in the dark. Considering the host species of antibodies used for co-staining experiments, sequential or simultaneous staining was performed. Coverslips were washed 3x in 1xPBS and mounted onto microscopy slides using Fluoromount-G with DAPI (Cat. No. 00-4959-52, Thermo Fisher Scientific). Z-stack images were acquired using the Leica SP8 Resonant Scanning Confocal (Leica Microsystems) using Leica Application Suite X (LAS X) version 3.5.7.23225 (Supplementary Fig. 4, 5), the Zeiss LSM 880 NLO fast Airyscan Confocal (Zeiss) using ZEN 2.3 SP1 software (Fig. 2f) or the DeltaVision OMX-SR system (GE Healthcare) using softWoRx 7.0.0 software and immersion oil with an index of refraction of 1.518 and an Olympus PlanApo oil immersion objective lens with 60x/1.42 NA (Fig. 5b, c). Microscopical images were analysed with Fiji software (version 2.1.0) and are presented as maximum intensity projections. Quantification analysis (Fig. 5c) was performed using Fiji software by creation of surface for KDEL and GM130 and subsequent calculation of spatial overlay resulting in the percentage of KDEL signal inside the Golgi.

**Patient PBMCs and FACS analysis**. Human blood samples were collected after informed consent (ethical review board of Istituto Giannina Gaslini-Genova-Italy N. BIOL 6/5/04). Healthy control (HC) samples were collected from adult donors at the blood donation centre of the Istituto Giannina Gaslini and anonymized for population characteristics.

PBMCs from a COPA syndrome patient, previously reported in[6] (blood sample collected at age 10 years), and two HC donors were isolated and treated with STING inhibitor H-151 (5 µM, InvivoGen) at 37 °C for 4 hrs. Cells were fixed with pre-warmed Fixation Buffer (BioLegend) at 37 °C for 15 min and permeabilized with pre-chilled True-Phos™ Perm Buffer (BioLegend) at $-20\,°C$ for 1 hr. Cells were then stained with CD3-APC mAb (SK7, BD Biosciences), CD14-FITC mAb (M5E2, BD Biosciences) and phospho-TBK1/NAK (Ser172) XP Rabbit mAb-PE (D52C2, Cell Signaling Technology) or Rabbit (DA1E) mAb IgG XP Isotype Control-PE (Cell Signaling Technology) at RT for 30 min. Samples were analyzed using the BD FACSCanto™ flow cytometer (BD Biosciences) and FlowJo 10.5 software.

The monocyte population was firstly identified based on cell size and granularity and subsequently confirmed by gating for the CD14-positive/CD3-negative subpopulation (Supplementary Fig. 6a). Data for phosphorylated TBK1 (pTBK1) is presented as histogram showing mean fluorescence intensity (MFI) for the CD14-expressing monocyte population. Quantification (Fig. 3b) shows fold change of MFI (representing pTBK1) after H-151 treatment and was calculated using this formula:

$$fold\ change(pTBK1) = MFI^{H-151} - MFI^{isotypectrl}/MFI^{untreated} - MFI^{isotypectrl}$$

**Statistical analysis**. Unless stated otherwise in the figure legends, data are presented as mean of independent biological replicates showing individual data points. Error bars represent standard error of the mean (SEM), unless otherwise stated. Using the GraphPad Prism 8 software, the statistical comparison was made either by unpaired Student's t-test, ratio paired Student's t-test or one-way ANOVA with subsequent multiple comparison testing as stated in the figure legends. P values are indicated by numbers or as * $P < 0.05$, ** $P < 0.01$, *** $P < 0.001$, **** $P < 0.0001$.

**Reporting summary**. Further information on research design is available in the Nature Research Reporting Summary linked to this article.

## Data availability

Data supporting the findings reported in this study are presented within the results section and the supplementary files. All graphs show individual data points pooled from replicate experiments. Raw data are available upon request from the corresponding author. The mass spectrometry proteomics data have been deposited to the ProteomeXchange Consortium via the PRIDE partner repository (http://proteomecentral.proteomexchange.org/cgi/GetDataset) with the dataset identifier PXD023135. Localization of identified STING-interacting proteins to ER or Golgi

compartments in Supplementary Data 1 was determined using the Gene Ontology database (https://www.ebi.ac.uk/QuickGO/, particularly cellular component terms). Other databases used include Uniprot (https://www.uniprot.org/) and LFQAnalyst (https://bioinformatics.erc.monash.edu/apps/LFQ-Analyst/). Source data are available with this paper for Figs. 1a, d, 2b, d, 3c-e, 4a, 5c, d, 6a, d and supplementary figures 1, 2a-c, 3, 7, 8a and 10.

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

## Acknowledgements

We thank Thomas Hayman (Melbourne University, WEHI) and Fiona Moghaddas (Melbourne University, WEHI) for kindly providing the single guide RNAs for MAVS and STING, respectively. We thank Cynthia Louis (Melbourne University, WEHI) for providing human *IL8* primers.

This work was supported by: Fellowships from the Australian National Health and Medical Research Council (NHMRC GNT2008699) (S.L.M.), HHMI-Wellcome International Research Scholarship 208694/Z/17/Z (S.L.M.), the Sylvia and Charles Viertel Foundation VTL2016F027 (S.L.M.), the National Health and Medical Research Council Early Career Fellowship (S.D. GNT1143412), the WEHI Centenary Fellowship (C.-H.Y.) and Ormond College's Thwaites Gutch Fellowship in Physiology (C.-H.Y.), Italian Ministry of Health, Ricerca Corrente (M.Ga.). M,Ge., G.H. and T.Z. received funding by the Deutsche Forschungsgemeinschaft (DFG, German Research Foundation) under Germany's Excellence Strategy – EXC2151 – 390873048 and TRR237 (G.H., T.Z.). A.S. is supported by the University of Melbourne through the International Research Training Program Scholarship and the DFG - GRK 2168. S.L.M. receives funding from Glaxosmithkline and IFM therapeutics. S.V. received financial support from the Italian Ministry of Foreign Affairs/Italian Health Ministry (PGR grant IN17GR10).

## Author contributions

A.S., K.H.S., S.D., I.P., C.H.Y., D.D.N., L.F.D., C.R.H., P.L., M.J.M. and S.V. performed or assisted with experimentation. A.S., S.D., I.P., K.H.S, C.H.Y., D.D.N., L.F.D., C.R.H., P.L., R.R.J.L., M.J.M., K.R.R., T.Z., G.H. M.P.G., S.V., M.Ga, M.Ge, and S.L.M. were involved in experimental analysis and interpretation. All authors contributed to the writing of this manuscript.

## Competing interests

M.Ga. declares consultancy and speakers fee from Novartis and SOBI. S.L.M. receives funding from IFM therapeutics and M.Ge. consults to IFM Therapeutics. The other authors declare no competing interests.
