## [Peer Review File · Nature Communications]

Deficiency in coatomer complex I causes aberrant activation of STING signallingREVIEWER COMMENTS

Reviewer #1 (Remarks to the Author):

In this manuscript, Steiner et al. describe in vitro models for COPA syndrome, as well as a role for STING in COPA syndrome. The authors show that COPA deficiency in THP-1 and HeLa cells leads to spontaneous induction of proinflammatory cytokines and type I interferon, which mimic the signatures observed in patients with COPA syndrome. Additionally, the authors report that deficiency in COPG, another subunit of COPI, also results in a similar inflammatory phenotype.

One major issue is that two other groups recently made similar reports, published in the Journal of Experimental Medicine. It seems that all three groups were simultaneously came to similar conclusions.

I have no concerns about the data or the rigor with which these experiments were performed.

I only have a couple of major questions and suggestions:

1) The authors mention a role for COPA in retrograde trafficking, but it is well-established that COPI vesicles also participate in intra-Golgi trafficking. The authors should consider this possibility as well. Is it possible that intra-Golgi anterograde trafficking is the real defect here? Or, alternatively, anterograde trafficking of STING for eventual degradation in the lysosome? Perhaps the possibility of intra-Golgi trafficking mediated by COPI vesicles should also be discussed?

2) One way to potentially increase novelty of this work would be to focus on another cell type. In animal models of STING-associated vasculopathy with onset in infancy and COPA syndrome, disease is mediated by T cells. Perhaps the authors could consider evaluating the role of the COPA mutant on STING signaling in T cells?

Reviewer #2 (Remarks to the Author):

The authors have investigated the molecular basis of immune dysregulation in COPA syndrome (caused by mutations in the COPA gene) using in vitro models for COPA syndrome and PBMCs from a COPA syndrome patient, and two healthy control (HC) donors. A range of sophisticated techniques have been employed to study signalling pathways in this rare autosomal dominant condition, and they have identified Stimulator of Interferon Genes (STING), as a key driver of inflammation.

Genetic deletion of COPG1, another COPI subunit protein, was associated with a type I IFN signature, and NF- κ B activation similar to COPA-deficiency. In vitro treatment of PBMCs with a small molecule STING inhibitor, H-151, produced a significant fold change of MFI (representing pTBK1) and immunoblotting reduced phosphorylation of IRF3 (pIRF3) and TBK1 (pTBK1).

They propose that defective retrograde transport in COPA syndrome interferes with STING trafficking and prolongs signalling, with increased transcription of proinflammatory cytokines, such as IL-6, and CXCL10 chemokine in addition to ISG mRNA expression levels reflecting a type I IFN signature.

General comment

The work is carefully planned and executed, with plausible data and interpretations. However, the conclusions are not entirely novel as quite similar results have already been published by "Deng et al. A defect in COPI-mediated transport of STING causes immune dysregulation in COPA syndrome. J Exp Med 2020;217(11):e20201045". So, to a significant extent these are confirmatory findings, largely observational, and represent incremental gains rather than significant breakthrough(s).

Reviewer #3 (Remarks to the Author):

The manuscript by Steiner et al explores the mechanistic basis for COPA syndrome. Using cells depleted of the COPI alpha (and G1) subunits, the authors recapitulate a signature transcriptional activation of proinflammatory cytokines and interferon stimulated genes that is characteristic of COPA syndrome. They defined the ER-Golgi recycling protein Stimulator of Interferon Genes (STING) as a mediator of the COPI-dependent isg activation response and recapitulated the role of STING in COPA syndrome patient-PBMCs by using a STING inhibitor.

It is noted that recent work detailed the role of STING activation through traffic and potential Golgi retention also in COPA syndrome models with some overall agreement. Deng et al linked COPI-interactions with a di lysine containing receptor (Surf4) that supports COPI-retrieval from the Golgi by retrograde traffic (a step that controls STING activation at the Golgi). Surf4 has also been identified in this work as an interactor of STING with no characterization. While the work is timely in that it provides support for the emerging roles of STING and traffic between the ER and the Golgi in COPA syndrome, some key aspects are preliminary in nature. In particular, the work falls short of defining traffic itineraries and roles of functional STING-COPI interactions in the process, which is key to understanding COPA syndrome.

1. An assumption that the authors make is that COPI depletion equals the mutations that characterize COPA syndrome. The authors use HeLa and THP-1 cells that are partially depleted of the alpha subunit of COPI (or gamma1, i.e. general disruption of the coat) as an experimental model system for COPA syndrome. However, the authors fail to define the functional outcomes of the depletions – and as a result fail to define the role of COPI in STING traffic or activation, how is it activated and where. We do not know if some subunits of the COPI complex are co-depleted (destabilized) under these conditions, if COPI is recruited to the Golgi under these conditions or if unassembled or misassembled COPI complex components inhibit general retrograde traffic (Fig. 1A). We do not know if under these conditions general traffic functionality is maintained with only selective rate limiting effects on STING. These questions should have been examined experimentally.

2. We do not have information on the precise localization of STING (and therefore the site of activation) under these COPI depletion conditions (Fig. 2F). Golgi disassembly under these conditions is implied by the authors yet is not examined. If it is the situation, could it be that activation of STING in this experimental model is derived from the mobilization of Golgi components into the ER? Does STING reside in the ER here? Is it in dispersed Golgi mini-stacks? or is it in ERGIC elements in the depleted cells? The quality of the images and overall analysis (Fig. 2F) is too low and organelle markers are lacking making it overall uninformative. In contrast to the authors model system of COPI depletion, expression of COPI alpha WD40 mutants inhibits interactions with Surf4 and arrests STING in a morphologically intact Golgi. The resulting localization of STING due to COPI depletion should be resolved experimentally using better morphological analysis and possibly cellular fractionation.

In a related point, the overexpression of WT COPI alpha subunit together with STING appears to activate signaling, which is slightly lower than that observed with overexpressed COPA syndrome mutants (Fig. 3C HEK293 cells). The results suggest that excess alpha subunit expression leads to inhibition of retrograde traffic. - The localization of STING and COPI as well as Golgi morphology under these conditions should be presented.

3. It would be useful to add all available information on the identified STING-interacting proteins (Fig. 2A). As it is, the presented plot does not provide much information. The authors do not use this analysis, which as it stands only serves to support data on Surf4-STING-COPI published elsewhere. It would be helpful at this point to highlight all of the detected STING interacting partners that are known to reside at the ER or in the Golgi and other proteins that may have functional interest. Analysis on the ability of selected ones to bind COPI (or COPII) would markedly improve the work.

4. The authors examined if depletion of COPII (Inhibition of COPII-mediated ER exit) activates STAT phosphorylation yet the logic here is unclear: activation of STING by COPA mutations requires that it first arrives from the ER to the ERGIC or Golgi. It has been shown that activation of STING by dsDNA is dependent on COPII mediated ER-exit (analyzed using the depletion of COPII inner layer Sar1a/b and Sec24 proteins). The authors finding that depletion of Sec13 does not activate STAT1 phosphorylation is therefore not really surprising. Interestingly, COPII-mediated ER exit of STING is

reported to be assisted by VPS34-PI3K activity. The utilization of PI3P may indicate aspects of autophagy rather than ER-Golgi traffic. The authors should analyze whether the co-depletion of COPII with COPA inhibits COPA-dependent activation as this experiment can address the site of STING activation in the author's COPA model (in case the Golgi is disassembled as implied by the authors) and the need for ER to Golgi traffic step here. Technically, it is noted that previous work by the Stephens group suggest that depletion of Sec13 (and a resulting loss of Sec31) fails to inhibit the traffic of a COPII-interacting reporter cargo from the ER. If this is the case here, the authors should target COPII inner layer proteins for depletion/analysis.

5. What is the reason for the decrease in Sec13 levels with H-151 treatment (Fig. 4C)? Does this COPII depletion contribute to the effects recorded with the drug in the analyzed COPA models? In other words, does this level of depletion sufficient to block STING arrival at the Golgi?

Manuscript NCOMMS-20-28744
Response to Reviewers

Reviewers' comments to the Authors:

Reviewer #1

In this manuscript, Steiner et al. describe *in vitro* models for COPA syndrome, as well as a role for STING in COPA syndrome. The authors show that COPA deficiency in THP-1 and HeLa cells leads to spontaneous induction of proinflammatory cytokines and type I interferon, which mimic the signatures observed in patients with COPA syndrome. Additionally, the authors report that deficiency in COPG, another subunit of COPI, also results in a similar inflammatory phenotype.

One major issue is that two other groups recently made similar reports, published in the Journal of Experimental Medicine. It seems that all three groups were simultaneously came to similar conclusions.

We appreciate this comment and are aware of the 3 other publications identifying STING as the main driver of type I Interferon response in COPA syndrome that were published by Deng et al., Lepelley et al. in the Journal of Experimental Medicine, Mukai et al. in Nature Communications (1-3) and one recently published additional report with complementary findings (4). Our manuscript was originally submitted around the same time as first 3 studies, however we experienced COVID-19 related issues that delayed the publication process. Indeed, the main conclusions drawn in our study support the findings by the other groups, however we use a different experimental approach employing CRISPR/Cas9-mediated genetic deletion of COPA in human cell lines to study effects of decreased COPA functionality. Furthermore, our study addresses the requirement of not only COPA, but also other subunits of the coatamer complex I (COPI) for STING regulation, generating an important addition to the current literature. Using *in vitro* models of COPA syndrome and patient PBMCs we demonstrate that inhibiting STING could be a promising approach for treatment of COPA syndrome. Further, we specifically identify COPI-mediated retrograde Golgi-to-ER but not intra-Golgi transport as the pathway involved in STING regulation.

I have no concerns about the data or the rigor with which these experiments were performed.

I only have a couple of major questions and suggestions:

1) The authors mention a role for COPA in retrograde trafficking, but it is well-established that COPI vesicles also participate in intra-Golgi trafficking. The authors should consider this possibility as well. Is it possible that intra-Golgi anterograde trafficking is the real defect here? Or, alternatively, anterograde trafficking of STING for eventual degradation in the lysosome? Perhaps the possibility of intra-Golgi trafficking mediated by COPI vesicles should also be discussed?

To address this question, we have used two different experimental approaches that examine whether STING signalling is regulated through COPI complex-mediated intra-Golgi trafficking.

The cytosolic Ca²⁺-dependent phospholipase A₂ alpha (cPLA2 α) is required for the formation of intercisternal tubules that connect Golgi stacks and are essential for intra-Golgi transport (5-7). Inhibition of cPLA2 α activity has been shown to suppress intra-Golgi trafficking (5). We hypothesized that if STING activation is caused by defective intra-Golgi trafficking, treatment of immortalized bone marrow-derived macrophages (iBMDMs) with cPLA2 α inhibitor AACOCF3 would induce spontaneous STING activation and result in phosphorylation of TBK1. However, we did not observe STING pathway activation tested with two different inhibitor concentrations indicating that defective intra-Golgi trafficking does not cause STING activation (**Rebuttal Figure 1A, Manuscript New Figure 6d**). To confirm that the cPLA2 α inhibitor was active at these concentrations, we evaluated expression levels of inducible nitric oxide synthase (iNOS) in response to IFN γ and LPS, which was significantly reduced (**Rebuttal Figure 1A, Manuscript New Figure 6d**) (8, 9).

The second approach involved the targeted deletion of 2 different COPG subunit isoforms that are distinctly enriched in cis-Golgi (COPG1, γ 1) or trans-Golgi (COPG2, γ 2) compartments (10, 11). This spatial segregation suggests different functions and specific roles of both isoforms with COPG1 more likely involved in mediating retrograde transport between Golgi and ER and COPG2 serving the intra-Golgi transport route (10). Taking advantage of this functional distinction, we deleted COPG1 and COPG2 isoforms individually in THP-1 cells using the CRISPR/Cas9 approach. Deletion efficiency was confirmed by immunoblotting for COPG1 and COPG2 transcript analysis by qRT-PCR due to the lack of a specific antibody (**Rebuttal Figure 1B,C, Manuscript New Figure 6a,b**). Interestingly, only deletion of COPG1 resulted in spontaneous phosphorylation of STAT1 and inflammatory gene transcription, whereas this was not observed in COPG2-deficient cells (**Rebuttal Figure 1B,D, Manuscript New Figure 6a,c**). Amelioration of inflammatory signalling in COPG1-deficient cells was achieved by treatment with STING inhibitor H-151 (**Rebuttal Figure 6A,B, Manuscript Figure 5d,e**), indicating STING pathway activation in COPG1-deficient THP-1 cells due to defective retrograde transport.

Collectively, these results demonstrate that deficiency in COPI-mediated retrograde transport and not intra-Golgi trafficking leads to subsequent dysregulation of STING homeostasis which is the relevant pathway underlying COPA syndrome pathology.

Rebuttal Figure 1: Targeted inhibition of anterograde intra-Golgi transport does not activate spontaneous STING signalling.

A) Intra-Golgi trafficking was inhibited in immortalized bone marrow-derived macrophages (iBMDM) by treatment with cPLA2 α inhibitor AACOCF3. Spontaneous activation of STING signalling was evaluated by immunoblotting for pTBK1, using ACTIN as a loading control. Inhibitor activity was analysed by its effect on iNOS expression levels following IFN γ priming (50 ng/ml overnight) and LPS stimulation (25 ng/ml, 6 hrs) in absence or presence of AACOCF3 (10 and 20 μ M, 30 min pre-incubation). A representative experiment of n=3 is shown. **B)** Targeted deletion of COPG1 and COPG2 subunits was performed in THP-1 cells using the CRISPR/Cas9 methodology. Immunoblotting for COPG1 confirmed successful deletion on protein level and levels of pSTAT1 indicate type I IFN pathway activation. A representative experiment of n=2 is shown. **C)** Reduction of COPG2 transcript levels in THP-1 cells (from B)) was confirmed by qRT-PCR analysis (n=2, error bars represent SD). **D)** qRT-PCR analysis of baseline inflammatory cytokine transcripts in THP-1 cells following targeted deletion of COPG1 and COPG2 subunits (n=2, error bars represent SD).

Furthermore, we did not observe significant changes in STING expression levels over the time period of Dox-induced COPA depletion, indicating that alteration of STING degradation via lysosomal and autophagy pathways is not a major factor in COPA^{deficient} THP-1 cells (**Rebuttal Figure 2., Manuscript New Supplementary Figure 10**).

Rebuttal Figure 2: Loss of COPA does not alter cellular expression levels of STING. Analysis of STING expression levels in THP-1 cell lysate over a time period of 96 hours (h) upon Doxycycline (Dox)-induced depletion of COPA. Representative of n=3 independent experiments.

2) One way to potentially increase novelty of this work would be to focus on another cell type. In animal models of STING-associated vasculopathy with onset in infancy and COPA syndrome, disease is mediated by T cells. Perhaps the authors could consider evaluating the role of the COPA mutant on STING signaling in T cells?

We analysed a panel of human immortalized T cells lines regarding endogenous cGAS and STING expression levels and found that although STING was ubiquitously expressed, cGAS was only detected in CEM, HuT-78 and PM1 cells (**Rebuttal Figure 3A**). Since HuT-78 cells showed high baseline levels of STAT1 phosphorylation (data not shown), we did not include this cell line in our analysis and used PM1 and CEM cells to investigate whether reduction of COPA levels is able to directly drive inflammatory signalling in T cell lines. CRISPR/Cas9-mediated gene deletion of COPA in PM1 cells was not able to consistently upregulate inflammatory signalling (**Rebuttal Figure 3B**), although Elsner et al. confirmed functionality of cGAS/STING signalling in this particular PM1 T cell line (12).

In CEM cells, reduction of COPA protein levels was associated with subtly increased transcription of *IFNB1* and type I IFN-induced genes and spontaneous phosphorylation of TBK1, which was consistent between experimental repeats (**Rebuttal Figure 3C,D**). Since STING signalling in T cells was previously shown to trigger pro-apoptotic gene expression, we analysed transcription levels of PUMA and NOXA, which were increased in COPA^{deficient} CEM cells (**Rebuttal Figure 3C**). These observations indicate that at least in some T cell lines, reduced expression levels of COPA and subsequent defective retrograde transport are able to induce subtle type I

IFN signalling, which is in line with inflammatory signalling observed in THP-1 and HeLa cells, and upregulation of proapoptotic genes, that has specifically been described as a STING-specific effect in T cells (13).

Rebuttal Figure 3: CRISPR/Cas9-mediated deletion of COPA induces type I IFN signalling in CEM but not in PM1 T cell lines. The CRISPR/Cas9 gene editing technology was employed to generate COPA^{deficient} human T cell lines using the sgRNA targeting exon 5 of the COPA gene. **A)** Western blot analysis of cGAS and STING protein expression levels in different immortalized human wildtype T cell lines. *n*=1 **B)** qRT-PCR analysis of type I IFN and ISGs in COPA-deleted PM1 cells. Data were pooled from *n*=4 independent experiments and are shown as mean ± SEM. Significance was assessed by unpaired Student's *t*-test. **C)** qRT-PCR analysis of IFNB1, ISGs, PUMA and NOXA baseline transcription levels in COPA^{deficient} CEM T cells. Data are pooled from *n*=5 independent experiments. Error bars represent SEM. Statistical significance was assessed by unpaired Student's *t*-test. **D)** Baseline protein expression levels of COPA and phosphorylated TBK1 (pTBK1) were assessed by immunoblotting, using ACTIN as a loading control. A representative result of *n*=3 experiments is shown. *P* values: ** *P*<0.01, *** *P*<0.001, *****P*<0.0001.

However surprisingly, pharmacological inhibition of this pathway using specific inhibitors targeting human STING (H-151) and cGAS (G140), did not ameliorate the observed phenotype (**Rebuttal Figure 4**). COPA^{deficient} CEM cells were treated with different concentrations of H-151 or G140 added for different time periods during Dox-induced COPA deletion (**Rebuttal Figure 4A,B**). Increased cytotoxicity was observed with H-151 at higher concentrations, therefore we also tested concentrations below 2.5 μM. Despite improved cell survival, type I IFN signalling was not reduced (**Rebuttal Figure 4C**). The functionality of H-151 was confirmed in THP-1 cells transfected with cGAS ligand HT DNA (**Rebuttal Figure 4D**).

Activation of cGAS/STING signalling in CEM WT cells was confirmed by electroporation with plasmid DNA, which induced a subtle increase in *TNF*, *IFNB1* and *ISG15* mRNA transcription after 7 hours incubation. Pre-incubation with a range of different concentrations of H-151 and G140 for 1 hour followed by plasmid DNA

electroporation was again not able to reduce inflammatory signalling in this experiment (**Rebuttal Figure 4E**). Furthermore, reduction of *IFNB1* and *ISG15* transcript levels was not achieved in DNA-electroporated CEM cells after shRNA-mediated knockdown of STING (**Rebuttal Figure 4F**). Multiple attempts to perform shRNA knockdown experiments in COPA^{deficient} CEM cells failed as the cell lines did not recover from puromycin selection.

Given that pharmacological inhibition and shRNA-mediated reduction of STING expression did not reduce DNA-induced inflammatory gene transcription in CEM T cells regardless of COPA expression levels, we concluded that this cell line does not provide a reliable model system to investigate cGAS/STING dependence of inflammatory signalling in COPA-deficiency. Nevertheless, deletion of COPA consistently induced increased inflammatory and apoptosis-inducing gene transcription in CEM T cells, which overall was more subtle compared to effects observed in THP-1 cell lines (main manuscript). However, whether inflammatory signalling occurs through activation of the cGAS/STING pathway, could not be conclusively assessed.

Previously, autoreactive T cells were shown to drive lung inflammation in a mouse model of COPA syndrome (14). This study identified inflammatory signalling in thymic epithelial cells expressing mutant COPA^{E241K/+} to have a detrimental effect on thymic T cell selection. As a consequence, increased numbers of lung autoreactive T cells and reduced numbers of regulatory T cells were observed in lungs from COPA^{E241K/+} mice (1) and in COPA^{V242G/+} mice (4). Our data here suggest that COPA-deficiency can directly induce aberrant type I IFN signalling in T cells, which may further contribute to disease progression. Whether this is dependent on STING mislocalisation, will need to be confirmed in other T cell lines or primary T cells and requires validation in a model system overexpressing COPA mutants, to exclude artefacts of the COPA deletion system as causing factors for this observation.

Rebuttal Figure 4: Parental and COPA^{deficient} CEM T cells were treated with STING inhibitor (H-151) and cGAS inhibitor (G140) (A, B). **A)** Cells were treated with STING inhibitor H-151, using DMSO as a vehicle control. Genetic deletion of COPA was induced by Doxycycline (Dox) treatment (1 μ g/ml) for 96 h which induced inflammatory gene transcription. The irreversible STING inhibitor H-151 (2.5 μ M, 5 μ M) was added 48 or 60 hours (h) after initiation of Dox treatment, incubated for 12 h (48-60 h) or 10 h (60-70 h), and washed away to reduce its cytotoxic effect at higher concentrations. Cells were harvested after 96 h incubation and

qRT-PCR analysis performed. **B)** CEM T cells were treated with 2.5 μM , 5 μM or 10 μM cGAS inhibitor G140 added 48 h or 72 h after start of Dox (1 $\mu\text{g}/\text{ml}$)-mediated COPA-deletion. Since no toxic effect was observed, cells were incubated with G140 until harvest after 96 h Dox incubation. **C)** To minimize the cellular toxicity, lower concentrations of H-151 (1 μM , 2.5 μM) were added 36 h after Dox addition, incubated for 12 h and washed off. Inflammatory transcript expression levels were analysed by qRT-PCR after a total incubation of 96 h. **D)** THP-1 WT cells were pre-incubated with DMSO vehicle control or H-151 at indicated concentrations and subsequently transfected with 2 $\mu\text{g}/\text{ml}$ HT DNA to induce cGAS/STING signalling. qRT-PCR analysis was performed after 7 h incubation. **E)** Electroporation of CEM WT cells with 24 μg plasmid DNA after cells were pre-incubated with indicated concentrations of H-151 and G140. After 7 hours incubation, transcription levels of TNF, IFNB1 and ISG15 were analysed by qRT-PCR. **F)** CEM parental cells were stably transduced with lentiviral supernatant carrying the pLKO.1 puro empty vector (EV, Addgene #8453) construct or pLKO.1 puro plasmid encoding a STING specific shRNA. After successful puromycin selection, DNA electroporation was performed using the same experimental procedure as described in E) and transcription levels of IFNB1 and ISG15 mRNA analysed by qRT-PCR after 7 h incubation. Reduction in STING levels was confirmed by qRT-PCR using specific primers. All experiments are $n=1$, technical duplicates are shown. Error bars indicate SD.

Additionally, we investigated inflammatory signalling in immortalized lung epithelial A549 cells since lung pathology is commonly observed in COPA syndrome patients. The results showed subtle upregulation of ISG transcription levels but not *IFNB1* following CRISPR/Cas9-induced sgRNA-mediated deletion of COPA (**Rebuttal Figure 5A**). Interestingly, we were not able to detect endogenously expressed STING by western blot (**Rebuttal Figure 5B**) which is also supported by findings of several other research groups (15) (16). Western blot analysis showed baseline phosphorylation of TBK1 in parental A549 cells, but an increase was not detectable by upon COPA deletion. These findings suggest that either below-detection expression levels of endogenous STING are sufficient to drive weak type I IFN signalling in A549 cells, or that activation of ISG transcription in this cell line occurs through pathways independent of STING.

Rebuttal Figure 5: CRISPR/Cas9-mediated COPA deletion in A549 lung epithelial cells induces subtle increase in ISG transcript levels (A) despite lack of endogenous STING expression (B). A) Results are pooled from $n=3$ independent experiments. B) Representative WB of $n=3$. Control THP-1 WT cells were treated with HT DNA (2 $\mu\text{g}/\text{ml}$) for 2 hours.

Whereas our findings in the main manuscript clearly demonstrate cGAS/STING dependence of inflammatory signalling observed in COPA^{deficient} THP-1 cells, we were not able to draw such conclusions for CEM T cells and A549 lung epithelial cells. This may reflect cell type-specific effects and suggest that COPA-deletion might be able to activate inflammatory signalling through STING-independent pathways in some cell types. Deletion of COPA also activates ER stress and the unfolded protein response (UPR), however no direct link to induction of type I IFN signalling has yet been established (17). However, ER stress, UPR pathways and subsequently induced NF- κ B signalling have been shown to influence production and sensing of type I IFN in various disease contexts (17). Interestingly, COPA and COPB subunits of the COPI complex were reported to be involved in transport and sorting of specific RNAs in neuronal cells (18). If defective, this dysregulation could perhaps contribute to activation of RNA-sensing pathways and drive cGAS/STING-independent type I IFN signalling, at least in some cell types.

In summary, we suggest that the investigation of pathways activated in COPA-deficient CEM and A549 cells is complicated, and beyond the scope of the current manuscript as it requires extensive validation.

Reviewer #2

The authors have investigated the molecular basis of immune dysregulation in COPA syndrome (caused by mutations in the COPA gene) using in vitro models for COPA syndrome and PBMCs from a COPA syndrome patient, and two healthy control (HC) donors. A range of sophisticated techniques have been employed to study signalling pathways in this rare autosomal dominant condition, and they have identified Stimulator of Interferon Genes (STING), as a key driver of inflammation. Genetic deletion of COPI1, another COPI subunit protein, was associated with a type I IFN signature, and NF- κ B activation similar to COPA-deficiency. In vitro treatment of PBMCs with a small molecule STING inhibitor, H-151, produced a significant fold change of MFI (representing pTBK1) and immunoblotting reduced phosphorylation of IRF3 (pIRF3) and TBK1 (pTBK1). They propose that defective retrograde transport in COPA syndrome interferes with STING trafficking and prolongs signalling, with increased transcription of proinflammatory cytokines, such as IL-6, and CXCL10 chemokine in addition to ISG mRNA expression levels reflecting a type I IFN signature.

General comment

The work is carefully planned and executed, with plausible data and interpretations. However, the conclusions are not entirely novel as quite similar results have already been published by “Deng et al. A defect in COPI-mediated transport of STING causes immune dysregulation in COPA syndrome. *J Exp Med* 2020;217(11):e20201045”. So, to a significant extent these are confirmatory findings, largely observational, and represent incremental gains rather than significant breakthrough(s).

We appreciate the concern, that has also been raised by reviewer 1. Please see our response to the general comment of reviewer 1.

Reviewer #3 (Remarks to the Author):

The manuscript by Steiner et al explores the mechanistic basis for COPA syndrome. Using cells depleted of the COPI alpha (and G1) subunits, the authors recapitulate a signature transcriptional activation of proinflammatory cytokines and interferon stimulated genes that is characteristic of COPA syndrome. They defined the ER-Golgi recycling protein Stimulator of Interferon Genes (STING) as a mediator of the COPI-dependent isg activation response and recapitulated the role of STING in COPA syndrome patient-PBMCs by using a STING inhibitor.

It is noted that recent work detailed the role of STING activation through traffic and potential Golgi retention also in COPA syndrome models with some overall agreement. Deng et al linked COPI-interactions with a di lysine containing receptor (Surf4) that supports COPI-retrieval from the Golgi by retrograde traffic (a step that controls STING activation at the Golgi). Surf4 has also been identified in this work as an interactor of STING with no characterization. While the work is timely in that it provides support for the emerging roles of STING and traffic between the ER and the Golgi in COPA syndrome, some key aspects are preliminary in nature. In particular, the work falls short of defining traffic itineraries and roles of functional STING-COPI interactions in the process, which is key to understanding COPA syndrome.

1. An assumption that the authors make is that COPI depletion equals the mutations that characterize COPA syndrome. The authors use HeLa and THP-1 cells that are partially depleted of the alpha subunit of COPI (or gamma1, i.e. general disruption of the coat) as an experimental model system for COPA syndrome. However, the authors fail to define the functional outcomes of the depletions – and as a result fail to define the role of COPI in STING traffic or activation, how is it activated and where.

We do not know if some subunits of the COPI complex are co-depleted (destabilized) under these conditions, if COPI is recruited to the Golgi under these conditions or if unassembled or misassembled COPI complex components inhibit general retrograde traffic (Fig. 1A).

We do not know if under these conditions general traffic functionality is maintained with only selective rate limiting effects on STING. These questions should have been examined experimentally.

We agree with this comment and have now included experiments to improve the characterization of our experimental model. The basis for this *in vitro* model system is the assumption that CRISPR/Cas9-mediated reduction of cellular COPA levels result in reduced functional retrograde transport that mimics the loss-of-function mutations described in COPA syndrome patients.

To address the question of whether depletion of COPA leads to destabilization or co-depletion of other COPI subunits, we performed immunoblot analysis of a selected subset of COPI subunits in COPA^{deficient} THP-1 cells (**Rebuttal Figure 6A, Manuscript New Figure 5d**). Furthermore, to investigate this question from a reverse angle, we deleted several other COPI subunits in THP-1 cells involved in formation of either subcomplex F (e.g. COPG1 (γ 1), COPD (δ)) or subcomplex B (COPE (ϵ)) and investigated subunit expression levels by immunoblot analysis (**Rebuttal Figure 6A, Manuscript New Figure 5d**). Unless intentionally deleted, only minor variability is observed for COPA, COPG1 and COPD protein expression levels across cell lines

deficient in different COPI subunits. This indicates that upon loss of COPA, CPG1 or COPD, mutual destabilization to some extent cannot be entirely excluded, however complete co-depletion of any of the here investigated subunits does not occur.

In contrast, COPE expression levels are markedly reduced in COPA^{deficient} cells, although COPA levels remain stable in cells deleted of COPE (**Rebuttal Figure 6A, Manuscript New Figure 5d**). One explanation for this is the direct interaction of subunits COPA (α) and COPE (ϵ) with each other and COPB1 (β') within the COPI subcomplex B (19-21). Experiments in yeast identified COPE as a structural component that stabilizes COPA, whilst being non-essential for normal retrograde trafficking (22). Therefore, the reduction of COPE levels upon COPA deletion is not surprising, however the underlying regulatory mechanisms are not yet understood. Interestingly, COPA stability does not seem impacted in COPE^{deficient} cells, indicating that retrograde transport likely remains intact in these cells.

In agreement with this is our finding that inflammatory pathway activation assessed by phosphorylation of STAT1 and increased transcription of *TNF*, *IFNB1* and type I IFN induced genes was observed in cells deficient for COPA, CPG1 and COPD but not COPE (**Rebuttal Figure 6A,B, Manuscript New Figure 5d,e**). Treatment with STING inhibitor H-151 was able to reduce proinflammatory signalling (**Rebuttal Figure 6A,B, Manuscript New Figure 5d,e**), suggesting that not only COPA but a functional COPI complex is required to regulate STING trafficking and signalling. In line with this, inflammatory signalling does not occur upon COPE deletion since functional retrograde trafficking and thus STING recycling is maintained in absence of this subunit (22).

However, this experiment solely investigated expression levels of COPI subunit proteins and no conclusion about complex assembly or functionality of incomplete COPI complexes can be drawn.

To confirm that retrograde transport is indeed impaired in the here used *in vitro* model of COPA-deficiency, we used immunofluorescence super-resolution microscopy and investigated localisation and intensity of KDEL staining in COPI-deficient HeLa cells. The amino acid sequence KDEL is an ER-specific retention signal present on soluble ER-resident proteins that are retrieved from Golgi to ER through COPI-mediated retrograde transport (23). We hypothesized that impaired retrograde transport would lead to accumulation of KDEL signal within Golgi compartments. As expected, we observed a significant shift in KDEL localisation towards Golgi compartments in COPA, CPG1 and COPD-deficient HeLa cells when compared to parental controls with intact retrograde transport (**Rebuttal Figure 6C,D, Manuscript New Figure 5b,c**). Quantification of this result is shown as the area ratio of KDEL signal overlaying with cis-Golgi marker GM130 indicating the percentage of KDEL signal inside the Golgi per cell (**Rebuttal Figure 6C, Manuscript New Figure 5c**).

Noteworthy is the overall increased KDEL signal intensity in COPA^{deficient} cells, which is likely due to the increased expression of ER chaperones (which carry the KDEL sequence) following ER stress and activation of the unfolded protein response (UPR) as a consequence of defective retrograde transport (24).

From these results we conclude that not only STING transport, but retrograde trafficking in general, is impaired upon deletion of COPA, COPG1 and COPD which is similar to the functional defect described for the COPA mutants.

Rebuttal Figure 6: CRISPR/Cas9-mediated deletion of COPI subunit proteins induces spontaneous type I IFN signalling in human monocytic THP-1 cells that is ameliorated by pharmacological inhibition of STING. A) Immunoblot analysis of pSTAT1 levels in THP-1 cells after CRISPR/Cas9-mediated genetic deletion of COPI-subunit proteins COPA, COPG1 (encoding subunit γ 1), COPD (encoding subunit δ) and COPE (encoding subunit ϵ) following treatment with STING inhibitor H-151 (2.5 μ M, 11 h). Cells were harvested after 72 h Dox treatment (1 μ g/ml). A representative of $n=3$ independent experiments is shown. **B)** qRT-PCR analysis of proinflammatory genes and ISGs of COPI-subunit deficient THP-1 cells similarly treated to A). Data are pooled from $n=3$ independent experiments and statistical significance was assessed by two-tailed ratio paired Student's t -test comparing Veh ctrl and H-151 treatment individually for each cell line. Error bars indicate SEM. **C)** Quantification of IF co-localization of KDEL and GM130 signal in COPI subunit-deficient HeLa cells. Results are

quantified as percentage area of KDEL localization (ER-specific retention signal) inside cis-Golgi (GM130). Each dot represents one single cell. Parental cells were combined for analysis. Statistical significance was assessed by one-way ANOVA, Line at median. **D)** Representative IF images of quantification shown in C) for COPA, COPG1 and COPD-deleted HeLa cells co-stained for KDEL (green), GM130 (magenta) and respective COPI subunit (cyan). P-values are indicated by numbers or as * $P < 0.05$, ** $P < 0.01$, *** $P < 0.001$, **** $P < 0.0001$.

2. We do not have information on the precise localization of STING (and therefore the site of activation) under these COPI depletion conditions (Fig. 2F). Golgi disassembly under these conditions is implied by the authors yet is not examined. If it is the situation, could it be that activation of STING in this experimental model is derived from the mobilization of Golgi components into the ER? Does STING reside in the ER here? Is it in dispersed Golgi mini-stacks? or is it in ERGIC elements in the depleted cells? The quality of the images and overall analysis (Fig. 2F) is too low and organelle markers are lacking making it overall uninformative. In contrast to the authors model system of COPI depletion, expression of COPI alpha WD40 mutants inhibits interactions with Surf4 and arrests STING in a morphologically intact Golgi. The resulting localization of STING due to COPI depletion should be resolved experimentally using better morphological analysis and possibly cellular fractionation.

Preliminary data on co-localization of STING-GFP with Golgi (GM130) and ER marker (KDEL) in HeLa cells stably reconstituted with STING-GFP have now been added to the manuscript and are shown here as **Rebuttal Figure 7 (Manuscript New Supplementary Figure 4)**. In this overexpression system, STING-GFP spontaneously formed multiple speckles in COPA^{deficient} cells (indicated by arrow) (**Rebuttal Figure 7A, Manuscript New Supplementary Figure 4a**), which largely co-localized with Golgi marker GM130, indicating STING-GFP localization to the dispersed Golgi (**Rebuttal Figure 7B, Manuscript New Supplementary Figure 4b**). No increased co-localization with ER marker (KDEL) was observed (**Rebuttal Figure 7C, Manuscript New Supplementary Figure 4c**).

In parental control cells, HT DNA-induced activation of STING-GFP triggered puncta formation at the Golgi (**Rebuttal Figure 7B, Manuscript New Supplementary Figure 4b**) but not the ER (**Rebuttal Figure 7C, Manuscript New Supplementary Figure 4c**). Therefore, despite the observed Golgi dispersal in COPA-depleted cells, this likely remains the site of STING accumulation and activation.

Rebuttal Figure 7: STING-GFP co-localizes with the dispersed Golgi in *COPA*^{deficient} HeLa cells stably expressing STING-GFP. sgRNA-mediated *COPA* deletion was induced by 72h doxycycline treatment. Parental and *COPA*^{deficient} HeLa-*STING*-GFP cells were fixed, permeabilized and stained for *COPA* (A), GM130 (B) or KDEL (C). Localization of activated *STING* in parental HeLa cells transfected with HT DNA (2 µg/ml, 2 h) is shown as positive control. Representative images, *n*=1, scale bar = 10 µm, arrows indicate *COPA*^{deficient} cells.

In a related point, the overexpression of WT COPI alpha subunit together with *STING* appears to activate signaling, which is slightly lower than that observed with overexpressed *COPA* syndrome mutants (Fig. 3C HEK293 cells). The results suggest that excess alpha subunit expression leads to inhibition of retrograde traffic. - The localization of *STING* and COPI as well as Golgi morphology under these conditions should be presented.

In our interpretation, the slightly increased pIRF3 and pTBK1 signal in HEK293T cells co-transfected with mCit-*STING* and WT *COPA* could be the result of an increased secretory workload caused by co-expression of *STING* with another protein in general, rather than an inhibitory effect of overexpressed WT *COPA* on retrograde transport. Transient transfection and high-level protein expression of mCit-*STING* alone (Manuscript Fig. 3C, mCit-*STING* + EV) induces spontaneous activation and phosphorylation of TBK1 above baseline levels. This is the result of high-level mRNA transcription under the EF1α promoter (pEF-BOS backbone), leading to high protein

levels of STING, spontaneous dimerization and Golgi translocation (25), a commonly observed disadvantage of this experimental setup. Additional overexpression of a second protein would further occupy ER folding capacity and secretory pathways, thereby limiting STING retrograde transport. From our own results and the recently published studies by Deng, Mukai, Lepelley et al. we now know that retrograde transport from Golgi to ER is essential to prevent spontaneous STING signalling. Therefore, in situations where protein translation overwhelms retrograde trafficking capacities, STING might accumulate at the Golgi which results in subsequent spontaneous activation. Therefore, we suspect that the increased pTBK1 and pIRF3 signals following co-transfection with COPA WT could be an artefact of increased ER workload, rather than an effect specific to WT COPA. Similarly, Kato et al. reported increased *Iffa4* and *Iffb1* luciferase reporter activity when WT COPA and hSTING were overexpressed in HEK293T cells, which was more pronounced in presence of COPA mutants (4).

3. It would be useful to add all available information on the identified STING-interacting proteins (Fig. 2A). As it is, the presented plot does not provide much information. The authors do not use this analysis, which as it stands only serves to support data on Surf4-STING-COPI published elsewhere. It would be helpful at this point to highlight all of the detected STING interacting partners that are known to reside at the ER or in the Golgi and other proteins that may have functional interest. Analysis on the ability of selected ones to bind COPI (or COPII) would markedly improve the work.

We appreciate this comment and have analysed the here identified STING-interacting proteins accordingly. The mass spectrometry proteomics data have been deposited to the ProteomeXchange Consortium via the PRIDE [1] partner repository with the dataset identifier PXD023135.

Cellular localisation of identified STING-interacting proteins to ER or Golgi compartments was determined using the Gene Ontology database (<https://www.ebi.ac.uk/QuickGO/>, particularly cellular component terms). The identified ER- and Golgi-resident proteins were allocated accordingly to two separate tabs in the attached excel sheet. Their ability to be trafficked by the COPI complex was analysed by identification of C-terminal COPI-binding motifs KDEL, RDEL, HDEL, KKxx, RKxx, KxKxx, KxHxx, HxHxx, KxRxx, RxKxx, RxR (26-30)(31-33). Although optimal ER retention through COPI strictly requires presence of di-lysine/di-arginine at position -3 and -4 or -3 and -5 relative to the C-terminus, Jackson et al. showed that presence of one or two lysines at various positions within the last 6 C-terminal residues possesses remaining ER retention ability, however to lesser extent than optimal binding motifs (28). Therefore, all C-terminal K, R and H residues of identified Golgi and ER-resident proteins were highlighted (bold) since binding to COPI complex cannot be excluded. However, only clearly identified COPI-binding motifs positioned at the above-mentioned residues were underlined and stated. Finally, identified proteins contributing to COPI/COPII complex formation were highlighted in green and previously identified STING interactors are highlighted in grey.

4. The authors examined if depletion of COPII (Inhibition of COPII-mediated ER exit) activates STAT phosphorylation yet the logic here is unclear: activation of STING by

COPA mutations requires that it first arrives from the ER to the ERGIC or Golgi. It has been shown that activation of STING by dsDNA is dependent on COPII mediated ER-exit (analyzed using the depletion of COPII inner layer Sar1a/b and Sec24 proteins). The authors finding that depletion of Sec13 does not activate STAT1 phosphorylation is therefore not really surprising. Interestingly, COPII-mediated ER exit of STING is reported to be assisted by VPS34-PI3K activity. The utilization of PI3P may indicate aspects of autophagy rather than ER-Golgi traffic. The authors should analyze whether the co-depletion of COPII with COPA inhibits COPA-dependent activation as this experiment can address the site of STING activation in the author's COPA model (in case the Golgi is disassembled as implied by the authors) and then need for ER to Golgi traffic step here. Technically, it is noted that previous work by the Stephens group suggest that depletion of Sec13 (and a resulting loss of Sec31) fails to inhibit the traffic of a COPII-interacting reporter cargo from the ER. If this is the case here, the authors should target COPII inner layer proteins for depletion/analysis.

This is a good point and we have investigated it further by deleting outer (SEC13) or inner (SEC24C) layer proteins of the COPII complex using CRISPR/Cas9 methodology in COPA^{deficient} THP-1 cells. Indeed, in our system, no difference in inflammatory phenotype between COPA^{deficient} and co-deleted COPA/SEC13^{deficient} THP-1 cells was observed as assessed by STAT1 phosphorylation and qRT-PCR analysis of *TNF*, *IFNB1* and several ISGs (**Rebuttal Figure 8A,B**).

In the case of inner layer COPII proteins, we co-deleted SEC24C in COPA^{deficient} THP-1 cells, since SEC24C was shown to directly interact with STING and adapter protein TMED2 (34) and knockdown of SEC24C in HeLa cells abrogated cGAMP-induced *IFNB1* transcription and STING translocation (35). However, the kinetics of co-deletion with COPA proved challenging, and we did not observe amelioration of STAT1 phosphorylation or target gene transcription (**Rebuttal Figure 8C,D**).

Potentially, this could be due to incomplete knockout efficiency or limitations of the inducible co-deletion system, since depending on the half-life of SEC24C and COPA proteins, STING accumulation at the Golgi might occur before levels of SEC24C are sufficiently reduced to prevent ER-to-Golgi STING transport.

Therefore, we have been unable to use our existing system to provide genetic evidence for a requirement of STING trafficking to the Golgi in COPA-deficiency, however we suggest that our improved imaging data (**Rebuttal Figure 7, Manuscript New Supplementary Figure 5**) provides some additional evidence for this.

Rebuttal Figure 8: Co-deletion of COPII subunits SEC13 or SEC24C did not impair STING signalling in COPA-deficiency. **A)** WB analysis of phosphorylated STAT1 (pSTAT1), SEC13 and COPA in COPA/SEC13^{deficient} THP-1 cell lines. **B)** qRT-PCR analysis of mRNA transcript levels of cell line used in A). Data were pooled from n=3 independent experiments, error bar represents SEM, statistical significance between COPA^{deficient} and COPA/SEC13^{deficient} cells was assessed using ratio-paired Student's t-test. *, P<0.05; ns, not significant. **C)** qRT-PCR analysis of SEC24C and proinflammatory gene transcript levels in THP-1 cells co-deleted for COPA and SEC24C. Technical duplicates of n=1 representative experiment are shown and presented as mean ± SD. **D)** Western blot analysis of cells used in C). All samples were run on the same gel but only lanes of interest are shown which required cropping of the blot. n=1.

5. What is the reason for the decrease in Sec13 levels with H-151 treatment (Fig. 4C)? Does this COPII depletion contribute to the effects recorded with the drug in the analyzed COPA models? In other words, does this level of depletion sufficient to block STING arrival at the Golgi?

We did not consistently observe a reduction of SEC13 levels with H-151 treatment in repeat experiments. Furthermore, results shown in **Rebuttal Figure 8** suggest that SEC13 is not required for STING trafficking to the Golgi (see response to comment 4). Therefore, the mechanism of H-151-mediated STING inhibition does not impact ER to Golgi transport by COPII.

References

1. Deng Z, Chong Z, Law CS, Mukai K, Ho FO, Martinu T, et al. A defect in COPI-mediated transport of STING causes immune dysregulation in COPA syndrome. *J Exp Med*. 2020;217(11).
2. Lepelley A, Martin-Nioclós MJ, Le Bihan M, Marsh JA, Uggenti C, Rice GI, et al. Mutations in COPA lead to abnormal trafficking of STING to the Golgi and interferon signaling. *J Exp Med*. 2020;217(11).
3. Mukai K, Ogawa E, Uematsu R, Kuchitsu Y, Kiku F, Uemura T, et al. Homeostatic regulation of STING by retrograde membrane traffic to the ER. *Nature Communications*. 2021;12(1):61.
4. Kato T, Yamamoto M, Honda Y, Orimo T, Sasaki I, Murakami K, et al. Augmentation of STING-induced type I interferon production in COPA syndrome. *Arthritis Rheumatol*. 2021.
5. San Pietro E, Capestrano M, Polishchuk EV, DiPentima A, Trucco A, Zizza P, et al. Group IV phospholipase A(2)alpha controls the formation of inter-cisternal continuities involved in intra-Golgi transport. *PLoS Biol*. 2009;7(9):e1000194.
6. de Figueiredo P, Drecktrah D, Katzenellenbogen JA, Strang M, Brown WJ. Evidence that phospholipase A₂ activity is required for Golgi complex and trans Golgi network membrane tubulation. *Proceedings of the National Academy of Sciences*. 1998;95(15):8642-7.
7. de Figueiredo P, Polizotto RS, Drecktrah D, Brown WJ. Membrane tubule-mediated reassembly and maintenance of the Golgi complex is disrupted by phospholipase A2 antagonists. *Mol Biol Cell*. 1999;10(6):1763-82.
8. Ng CY, Kannan S, Chen YJ, Tan FCK, Ong WY, Go ML, et al. A New Generation of Arachidonic Acid Analogues as Potential Neurological Agent Targeting Cytosolic Phospholipase A2. *Scientific Reports*. 2017;7(1):13683.
9. Paloschi MV, Lopes JA, Boeno CN, Silva MDS, Evangelista JR, Pontes AS, et al. Cytosolic phospholipase A2- α participates in lipid body formation and PGE2 release in human neutrophils stimulated with an l-amino acid oxidase from *Calloselasma rhodostoma* venom. *Scientific Reports*. 2020;10(1):10976.
10. Moelleken J, Malsam J, Betts MJ, Movafeghi A, Reckmann I, Meissner I, et al. Differential localization of coatamer complex isoforms within the Golgi apparatus. *Proc Natl Acad Sci U S A*. 2007;104(11):4425-30.
11. Wegmann D, Hess P, Baier C, Wieland FT, Reinhard C. Novel Isotypic γ/ζ Subunits Reveal Three Coatamer Complexes in Mammals. *Molecular and Cellular Biology*. 2004;24(3):1070-80.
12. Elsner C, Ponnurangam A, Kazmierski J, Zillinger T, Jansen J, Todt D, et al. Absence of cGAS-mediated type I IFN responses in HIV-1-infected T cells. *Proceedings of the National Academy of Sciences*. 2020;117(32):19475-86.
13. Gulen MF, Koch U, Haag SM, Schuler F, Apetoh L, Villunger A, et al. Signalling strength determines proapoptotic functions of STING. *Nature Communications*. 2017;8(1):427.
14. Deng Z, Law CS, Ho FO, Wang KM, Jones KD, Shin J-S, et al. A Defect in Thymic Tolerance Causes T Cell-Mediated Autoimmunity in a Murine Model of COPA Syndrome. *The Journal of Immunology*. 2020;2000028.
15. Olagnier D, Brandtoft AM, Gunderstofte C, Villadsen NL, Krapp C, Thielke AL, et al. Nrf2 negatively regulates STING indicating a link between antiviral sensing and metabolic reprogramming. *Nature Communications*. 2018;9(1):3506.

16. Massa D, Baran M, Bengoechea JA, Bowie AG. PYHIN1 regulates pro-inflammatory cytokine induction rather than innate immune DNA sensing in airway epithelial cells. *J Biol Chem*. 2020;295(14):4438-50.
17. Sprouten J, Garg AD. Type I interferons and endoplasmic reticulum stress in health and disease. *Int Rev Cell Mol Biol*. 2020;350:63-118.
18. Todd AG, Lin H, Ebert AD, Liu Y, Androphy EJ. COPI transport complexes bind to specific RNAs in neuronal cells. *Hum Mol Genet*. 2013;22(4):729-36.
19. Lowe M, Kreis TE. In vitro assembly and disassembly of coatomer. *J Biol Chem*. 1995;270(52):31364-71.
20. Lowe M, Kreis TE. In vivo assembly of coatomer, the COP-I coat precursor. *J Biol Chem*. 1996;271(48):30725-30.
21. Fiedler K, Veit M, Stamnes MA, Rothman JE. Bimodal Interaction of Coatomer with the p24 Family of Putative Cargo Receptors. *Science*. 1996;273(5280):1396-9.
22. Duden R, Kajikawa L, Wuestehube L, Schekman R. ϵ -COP is a structural component of coatomer that functions to stabilize α -COP. *The EMBO Journal*. 1998;17(4):985-95.
23. Munro S, Pelham HR. A C-terminal signal prevents secretion of luminal ER proteins. *Cell*. 1987;48(5):899-907.
24. Watkin LB, Jessen B, Wiszniewski W, Vece TJ, Jan M, Sha Y, et al. COPA mutations impair ER-Golgi transport and cause hereditary autoimmune-mediated lung disease and arthritis. *Nat Genet*. 2015;47(6):654-60.
25. Ergun SL, Fernandez D, Weiss TM, Li L. STING Polymer Structure Reveals Mechanisms for Activation, Hyperactivation, and Inhibition. *Cell*. 2019;178(2):290-301.e10.
26. Ma W, Goldberg J. Rules for the recognition of dilysine retrieval motifs by coatomer. *Embo j*. 2013;32(7):926-37.
27. Szul T, Sztul E. COPII and COPI Traffic at the ER-Golgi Interface. *Physiology*. 2011;26(5):348-64.
28. Jackson MR, Nilsson T, Peterson PA. Identification of a consensus motif for retention of transmembrane proteins in the endoplasmic reticulum. *Embo j*. 1990;9(10):3153-62.
29. Nilsson T, Jackson M, Peterson PA. Short cytoplasmic sequences serve as retention signals for transmembrane proteins in the endoplasmic reticulum. *Cell*. 1989;58(4):707-18.
30. Burman JL, Bourbonniere L, Philie J, Stroh T, Dejgaard SY, Presley JF, et al. Scyl1, mutated in a recessive form of spinocerebellar neurodegeneration, regulates COPI-mediated retrograde traffic. *J Biol Chem*. 2008;283(33):22774-86.
31. Roth D, Lynes E, Riemer J, Hansen Henning G, Althaus N, Simmen T, et al. A di-arginine motif contributes to the ER localization of the type I transmembrane ER oxidoreductase TMX4. *Biochemical Journal*. 2009;425(1):195-208.
32. Newstead S, Barr F. Molecular basis for KDEL-mediated retrieval of escaped ER-resident proteins – SWEET talking the COPs. *Journal of cell science*. 2020;133(19).
33. Anelli T, Alessio M, Bachi A, Bergamelli L, Bertoli G, Camerini S, et al. Thiol-mediated protein retention in the endoplasmic reticulum: the role of ERp44. *The EMBO Journal*. 2003;22(19):5015-22.
34. Sun M-S, Zhang J, Jiang L-Q, Pan Y-X, Tan J-Y, Yu F, et al. TMED2 Potentiates Cellular IFN Responses to DNA Viruses by Reinforcing MITA Dimerization and Facilitating Its Trafficking. *Cell Reports*. 2018;25(11):3086-98.e3.

35. Gui X, Yang H, Li T, Tan X, Shi P, Li M, et al. Autophagy induction via STING trafficking is a primordial function of the cGAS pathway. *Nature*. 2019;567(7747):262-6.

REVIEWER COMMENTS

Reviewer #2 (Remarks to the Author):

The authors have employed a wide range of laboratory-based techniques to identify COPI-mediated retrograde Golgi-to-ER transport as a key pathway in STING regulation. An impressive amount of new data has been generated to support defective COPI-mediated retrograde Golgi-to-ER trafficking, with aberrant regulation of STING, via cGAS activation, and associated inflammatory cytokine release and type I interferon-driven inflammation as some of the major defects in COPA syndrome. They show that other subunits of the coatamer complex I (COPI), apart from COPA, namely COPG1 and COPD, are also involved in STING regulation.

The regulation of ER to Golgi secretory pathways and trafficking through the complex is incompletely understood, and this thought-provoking study offers fresh insights into the whole system. The authors acknowledge certain limitations of their in vitro model systems and I have no major disagreements with the conclusions on that basis.

They also studied PBMCs from a COPA syndrome patient carrying the c.698G>A (R233H) mutation, which revealed raised levels of pTBK1 in CD14- expressing monocytes. There was a drop in pTBK1 levels after H-151 treatment (by MFI), suggesting basal STING activation in the patient's PBMCs.

Furthermore, this study raises the possibility of other immune-mediated disorders being associated with mutations in other COPI subunit proteins, with the prospect of using cGAS/STING inhibition as a therapeutic option in COPA syndrome and other trafficking disorders.

Michael McDermott (Leeds)

Reviewer #3 (Remarks to the Author):

In their original submission, Steiner et al showed that defects in the retrograde traffic of STING leads to activation in COPA syndrome. Several studies published at the time defined the role of COPA mutations, and SURF4, in this retrograde traffic step, inhibition of which leads to STING activation. Clear interpretation of the authors original data that used a COPI depletion strategy (rather than mutated COPA) was supported by these other studies. In their revised manuscript, the authors added extensive work to address my concerns, mainly to show that depletion of other COPI subunits leads to general defects in retrograde traffic that similarly correlate with STING activation (and thus are equivalent to mutations found in COPA patients). However, given the authors depletion strategy and its effects on the integrity of the Golgi, localization of activated STING still remains unclear but the model agrees with published data. Importantly, the authors move beyond these conclusions mainly by adding data on selective usage of COPG1 (and retrograde traffic) in STING activation vs COPG2 (perhaps intra Golgi traffic anterograde traffic). They provide some pharmacology to support this notion. The added data are interesting yet preliminary, comparing mRNA to protein levels to assign presumed anterograde or retrograde roles, lack of controls for the affected traffic routes or mechanism for the arrival of STING at the TGN. The use of a PLA2 inhibitor with a similar goal is less specific and may affect other traffic routes. The revised study is improved and certainly adds to our understanding of COPA syndrome and the roles COPI and cGAS in STING activation however it remains confirmatory to several published papers, while the added data are preliminary.

Manuscript NCOMMS-20-28744
Point-by-point response to the REVIEWERS' COMMENTS

Reviewer #2 (Remarks to the Author):

The authors have employed a wide range of laboratory-based techniques to identify COPI-mediated retrograde Golgi-to-ER transport as a key pathway in STING regulation. An impressive amount of new data has been generated to support defective COPI-mediated retrograde Golgi-to-ER trafficking, with aberrant regulation of STING, via cGAS activation, and associated inflammatory cytokine release and type I interferon-driven inflammation as some of the major defects in COPA syndrome. They show that other subunits of the coatamer complex I (COPI), apart from COPA, namely COPG1 and COPD, are also involved in STING regulation.

The regulation of ER to Golgi secretory pathways and trafficking through the complex is incompletely understood, and this thought-provoking study offers fresh insights into the whole system. The authors acknowledge certain limitations of their in vitro model systems and I have no major disagreements with the conclusions on that basis.

They also studied PBMCs from a COPA syndrome patient carrying the c.698G>A (R233H) mutation, which revealed raised levels of pTBK1 in CD14- expressing monocytes. There was a drop in pTBK1 levels after H-151 treatment (by MFI), suggesting basal STING activation in the patient's PBMCs.

Furthermore, this study raises the possibility of other immune-mediated disorders being associated with mutations in other COPI subunit proteins, with the prospect of using cGAS/STING inhibition as a therapeutic option in COPA syndrome and other trafficking disorders.

Michael McDermott (Leeds)

Reviewer #3 (Remarks to the Author):

In their original submission, Steiner et al showed that defects in the retrograde traffic of STING leads to activation in COPA syndrome. Several studies published at the time defined the role of COPA mutations, and SURF4, in this retrograde traffic step, inhibition of which leads to STING activation. Clear interpretation of the authors original data that used a COPI depletion strategy (rather than mutated COPA) was supported by these other studies.

In their revised manuscript, the authors added extensive work to address my concerns, mainly to show that depletion of other COPI subunits leads to general defects in retrograde traffic that similarly correlate with STING activation (and thus are equivalent to mutations found in COPA patients). However, given the authors depletion strategy and its effects on the integrity of the Golgi, localization of activated STING still remains unclear but the model agrees with published data.

This concern of the reviewer is already addressed as follows in the discussion of the manuscript:

'STING signalling in COPA-depleted HeLa cells therefore may occur from smaller Golgi fragments, rather than an intact Golgi complex. In line with our data, spontaneous Golgi accumulation of STING in cell lines overexpressing COPA mutants and COPA syndrome patient cells has recently been shown in other studies published while our manuscript was under review (53, 54, 81). Since Golgi dispersal was not observed in the complementary studies (9, 53, 54, 81), this is likely a distinct feature of the COPA-deletion model system. Importantly, ligand-stimulated STING activation is retained in our COPA^{deficient} cells, suggesting that the observed Golgi fragmentation does not impair STING signalling (84).'

Importantly, the authors move beyond these conclusions mainly by adding data on selective usage of COPG1 (and retrograde traffic) in STING activation vs COPG2 (perhaps intra Golgi traffic anterograde traffic). They provide some pharmacology to support this notion. The added data are interesting yet preliminary, comparing mRNA to protein levels to assign presumed anterograde or retrograde roles, lack of controls for the affected traffic routes or mechanism for the arrival of STING at the TGN.

The limitations using the presumed selective roles of COPG1 and COPG2 have been mentioned in the following paragraph:

'Despite the described differences in localization of COPG1- and COPG2-containing COPI vesicles within secretory pathway compartments (71), the suggested functional difference between both paralogues has not yet been confirmed experimentally (72). Proteomic profiling of $\gamma 1\zeta 1$, $\gamma 1\zeta 2$, $\gamma 2\zeta 1$ COPI vesicle contents did not reveal significant differences (73), however mediation of specific transport directions has not been investigated.'

The use of a PLA2 inhibitor with a similar goal is less specific and may affect other traffic routes.

The possibility of off-target effects in this experiment has been added to the results (page 14).

The revised study is improved and certainly adds to our understanding of COPA syndrome and the roles COPI and cGAS in STING activation however it remains confirmatory to several published papers, while the added data are preliminary.